

# Spatial variations of deep soil moisture and the influencing factors in the Loess Plateau, China

**X. N. Fang[1], W. W. Zhao[1], L. X. Wang [2], Q. Feng [3], J. Y. Ding [1], Y. X. Liu [1] and X. Zhang [1]**

[1] State Key Laboratory of Earth Surface Processes and Resource Ecology, College of Resource and Technology, Beijing Normal University, Beijing 100875, P. R. China

[2] Department of Earth Sciences, Indiana University-Purdue University, Indianapolis (IUPUI), Indianapolis, Indiana 46202, USA

[3] College of Forestry, Shanxi Agricultural University, Taigu, Shanxi 030801, P. R. China

Correspondence to: W. W. Zhao (Zhaoww@bnu.edu.cn) and L. X. Wang (lxwang@iupui.edu)





## Abstract:

Soil moisture in deep soil layers is a relatively stable water resource for vegetation growth in the semi-arid Loess Plateau of China. Characterizing the spatial variations of deep soil moisture and its influencing factors at a moderate watershed scale is important to ensure the sustainability of vegetation restoration efforts. In this study, we focused on analyzing the spatial variation and factors influencing soil moisture content (SMC) in (0-500 cm) soil layers based on a soil moisture survey of the Ansai watershed, Yanan, Shannxi province. Our results can be divided into four main findings. (1) At the watershed scale, the higher spatial variation of deep SMC occurred at 0-20 cm, 120-140 cm and 480-500 cm in the vertical direction. At a comparable depth but in the horizontal direction, the spatial variation of deep SMC under native vegetation was much lower than that in human-managed vegetation and introduced vegetation. (2) The deep SMC in native vegetation and human-managed vegetation was significantly higher than that of introduced vegetation, and different degrees of soil desiccation occurred under all introduced vegetation types. (3) Taking the SMC condition of native vegetation as a reference for local control, soil could be divided into four layers: I) shallow rapid change layer (0-60 cm); II) main rainfall infiltration layer (60-220 cm); III) transition layer (220-400 cm); and IV) stable layer (400-500 cm). Positive and significant correlations existed between SMC at layers II, III and IV, and the correlations of the neighboring layer ranges were clearly stronger than that of nonadjacent depth ranges, although the SMC at shallow rapid change layer I showed a disconnect (i.e., no correlations) with those at the three other soil depth layers. (4) The influencing factors of deep SMC at the watershed scale varied with land management types. The main local controls of SMC variation were soil particle composition and annual average rainfall; human agricultural management measures can alter soil buck density, which contributes to higher deep SMC. In introduced vegetation, plant growth conditions, planting density, and litter water



holding traits showed significant relationships with deep SMC. The results of this
study are of practical significance for vegetation restoration strategies and the
sustainability of restored ecosystems.

**1 Introduction**

Soil moisture is an indispensable component of the terrestrial system and plays
a critical role in surface hydrological processes, especially runoff generation, soil
evaporation and plant transpiration (Baroni et al., 2013; Cheema et al., 2011; Chen et
al., 2008a; Chen et al., 2007; Legates et al., 2010; Sun et al., 2015; Wang et al.,
2012a; Wang et al., 2015b; Zhang et al., 2014; Zhao et al., 2013). Moisture in deep
soil layers is essential and is closely connected to shallow soil moisture and deep
groundwater. It also works as a reservoir for soil, which is important for plant growth
in dry seasons (Jia and Shao, 2014; Yang et al., 2012b). This is particularly true in
semi-arid areas, such as the Loess Plateau of China, where water resources are
incredibly scarce. In such regions, deep soil moisture even becomes the main
constraining factor of plant productivity and ecosystem sustainability (Wang et al.,
2011a; Wang et al., 2010b).
The Loess Plateau of China is located in a semi-arid area. The average annual
rainfall in this region ranges from 150 to 800 mm, which is far lower than the
average annual pan evaporation (1400–2000 mm) (Wang et al., 2010a). Low
precipitation and high evaporation results in lower soil moisture content in this
region. The shallow soil moisture is not sufficient to meet the needs of introduced
vegetation growth. Moreover, loess soil thickness in this area ranges from 30-80 m;
at these depths, groundwater is not available for plants (Wang et al., 2013).
Therefore, deep soil moisture, which is stored in unsaturated soil, becomes an
important water resource for plant growth (Yang et al., 2012b). However, the
vegetation introduced by the national Grain for Green project tends to have strong
water consumption. Large-scale afforestation has resulted in the excessive
consumption of deep soil moisture, and a large range of soil desiccation has been
reported (Wang et al., 2008b; Wang et al., 2010a; Wang et al., 2010b; Wang et al.,





2011b). Soil desiccation greatly reduces the capability of a "soil reservoir" to supply

water to deep soil layers for plant growth in the Loess Plateau (Chen et al., 2008b).

Introduced vegetation in desiccated land is easily degraded with low productivity,

and "small aged tree" with a height of 3–5 m appeared widely. Therefore the

sustainability of the restored ecosystem is being challenged. Moreover, traditional

soil moisture studies, which have mainly focused on shallow depth layers (Baroni et

al., 2013; Bi et al., 2009; Gómez-Plaza et al., 2001), clearly cannot reveal the

sustainability need for vegetation restoration.

Studies on deep soil moisture have gradually drawn attention from many

scientists in recent years. For example, it was recently found that deep soil moisture

was excessively consumed by almost all the introduced vegetation, and high planting

density was the main reason for the severe deficit of soil moisture (Yang et al.,

2012b). It was also found that introduced vegetation diminished the spatial

heterogeneity of deep soil moisture at the small catchment scale (Jia and Shao, 2014;

Yang et al., 2014b). In recent years, several studies have been conducted on the

spatial variation and influencing factors of deep soil moisture in the Loess Plateau

(Jia and Shao, 2014; Liu et al., 2010; Sun et al., 2014; Wang et al., 2012b; Wang et

al., 2013; Yang et al., 2014a). Deep soil moisture is an indispensable water source for

vegetation growth in the semi-arid Loess Plateau; understanding the spatial variation

and influencing factors of deep soil moisture is important for "timely, suitable, and

moderate" vegetation restoration, and it can also help in developing proper measures

that can help control soil desiccation. In fact, deep soil moisture content (SMC) is a

result of long-term biophysical processes controlled by multiple factors (Vereecken

et al., 2007). Several factors may impact soil moisture variation, such as vegetation

traits, soil properties, topographical factors, climate factors, and human landscape

management measures (Lu et al., 2007; Lu et al., 2008; Montenegro and Ragab,

2012; Qiu et al., 2001; Vivoni et al., 2008; Zhu and Lin, 2011). The dominant factors

that affect deep SMC spatial variation depend on the research scale (Entin et al.,

2000). For instance, deep SMC variation was found to be mainly dominated by the





type of vegetation at the slope scale (0.1-1 km$^2$) (Jia et al., 2013). It was also found
that vegetation and topography are key factors contributing to deep SMC variation at
the small catchment scale (1-100 km$^2$) (Yang et al., 2012a). Meanwhile, Wang et al.
(2012b) reported that deep SMC variation at the regional scale (i.e., the whole Loess
Plateau, covering 640,000 km$^2$) is mainly determined by plant types and climatic
conditions. Note that vegetation factors play an important role in the spatial variation
of deep SMC at all scales (Western et al., 2004). While all spatial scales, from slopes
and small catchments to regions are relevant to the understanding of deep SMC
variation, some scales are more operational and meaningful than others. For example,
slopes and small catchments based studies tend to be too small in spatial extent to
incorporate all environmental factors and human-managed measures (soil traits,
climate characteristics, and human-managed measures in one slope or small
catchment are usually homogeneous) that most relevant to deep SMC variation (Bi et
al., 2009; Gόmez‑Plaza et al., 2000; Zhu et al., 2014a; Zhu et al., 2014b), whereas
at the region scale, it is often impossible to assess essential mechanistic details (high
variation of rainfall and temperature can cover the influencing effects of other
factors) of deep SMC variation necessary for guiding local policies (Wang et
al.,2010a; Wang et al., 2010b; Wang et al., 2012). A moderate scale, covering an area
of approximately 100-1000 km$^2$ over a watershed or a geopolitically-defined area
represents a pivotal scale domain for the research of deep SMC variation mechanism.
In particular, it is the scale at which people and nature mesh and interact most
acutely (Fang et al., 2015; Zhao and Fang, 2014), and thus is a more operational
scale for sustainable vegetation restoration policy making. Up to date, however, little
particular research of deep SMC variation has centered on such a moderate scale,
and the variation mechanism of deep SMC at this kind of scale is still unclear.
In this study, we aimed to reveal the spatial variation of deep SMC and its
influencing factors at a moderate watershed scale. According to previous studies,
factors that control deep SMC variations are different under three land management
types: native vegetation with a shallow root system, introduced vegetation with a





deep root system, and vegetation with agricultural management measures (Jia et al., 2013; Jia and Shao, 2014; Yang et al., 2012b; Yang et al., 2014a). This study included all three land management types, covering eight specific vegetation types. We first explored the overall variation of SMC in this area and then compared the deep SMC of these three different land management types as well as identify variations in their profiles. Furthermore, the influence of various environmental factors on deep SMC under different vegetation types is discussed. The objectives of this study were to: (1) quantify the spatial variation characteristics of deep SMC; (2) explore the mechanisms for controlling deep SMC variability among different land management types at the watershed scale; (3) develop recommendations for land use management and the sustainability of vegetation recovery for the Loess Plateau.

## 2 Materials and Methods

### 2.1 Study area

The Yanhe watershed lies in the middle of the Loess Plateau in the northern Shaanxi Province. The Ansai watershed (108°47′-109°25′E, 36°52′-37°19′N) (Fig. 1) in this study is located in the upstream section of the Yanhe river, covering an area of approximately 1334 km$^2$, with a highly fragmented terrain; the elevation here ranges from 1057 m to 1743 m above sea level. This typical semi-arid loess hilly region has a mean annual temperature of 8.8 ℃ and an average annual precipitation of 505 mm. Most rainfall occurs in the form of thunderstorms during the summer months from July to September. Soil types in this study area include mainly loess soil with low fertility and vulnerability to soil erosion (Zhao et al., 2012).

The Ansai watershed is located on a warm forest steppe; the predominant land use types in the watershed are rain-fed farmland, orchard land, sparse native grassland, pasture grassland, shrub land, and forest (Feng et al., 2013). The native vegetation in the study area consists of sparse grasses with shallow roots dominated by species, such as bunge needlegrass, common leymus, and Altai heterpappus. Non-native species, such as alfalfa, black locust, David peach, sea buckthorn, and *Caragana korshinskii*, were predominantly used in the study area under the national





Grain for Green project. The cultivated crops are predominantly maize, millet and
broom corn millet. Being in a semi-arid climatic zone, water resources represent the
major constraint of vegetation growth and agricultural crop production.

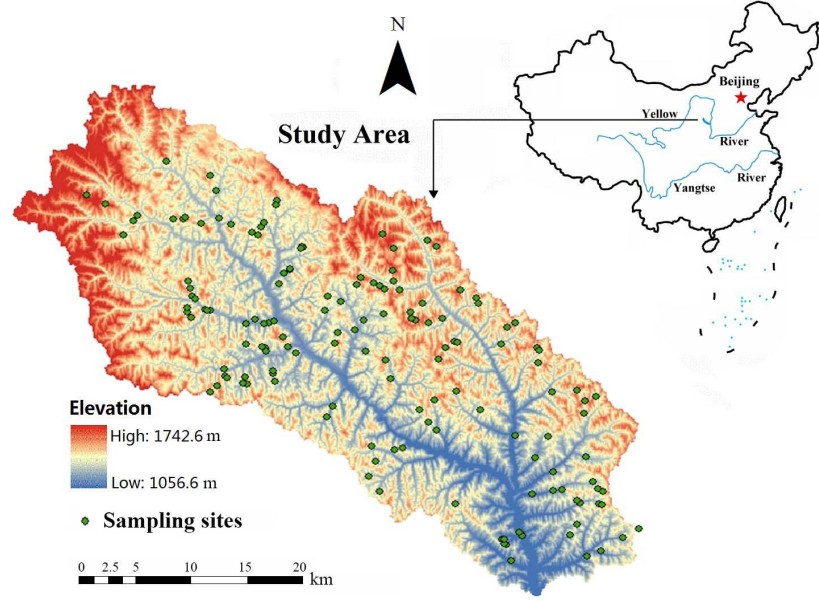

Figure 1. Location of the study area and sampling sites.

## 2.2 Sampling locations and description

In this study, three land management types were selected, including: (1) native
shallow root vegetation: native grasses (NG); (2) introduced deep root vegetation:
pasture grasses (PG), sea buckthorn (SB), *Caragana korshinskii* (CK), David peach
(DP), and black locust (BL); (3) human-managed vegetation: farmland (FL) and
apple orchard (AO). To fully explore the influencing factors of deep soil moisture,
we identified the following four types of factors: topography factors, soil properties,
vegetation traits, and climate factors, which further included 23 independent
variables: average annual rainfall (AAR), altitude (Al), slope position (SP), slope
aspect (SA), slope gradient (SG), clay (Cl), silt (Sl), sand (Sa), organic (Or), porosity
(Po), soil bulk density (SBD), vegetation coverage (VC), grass biomass (GB), grass
height (GH), planting density (PD), plant height (PH), diameter at breast height





(DBH), crown width (CW), basal diameter (BD), litter max water holding (LMWH),
litter biomass (LB), and clear bole height (CBH). The distance between each
vegetation sampling site was at least 2 km. The sampling locations are shown in Fig.
1. The main characteristics and sampling numbers for each vegetation type are
shown in Table 1.
Table 1. Main characteristics and sampling numbers for different vegetation types.

| Vegetation conditions | Native vegetation | Managed vegetation | | Introduced vegetation | | | | |
|---|---|---|---|---|---|---|---|---|
| | NG [a] | FL | AO | PG | CK | SB | DP | BL |
| Sampling number | 25 | 22 | 10 | 11 | 18 | 15 | 12 | 38 |
| Altitude (m) | 1392.60 | 1380.14 | 1370.10 | 1401.00 | 1350.61 | 1435.67 | 1377.58 | 1326.54 |
| Slope aspect (°) | 170.67 | 200.60 | 173.5 | 195.43 | 161.75 | 195.77 | 128.09 | 156.36 |
| Slope gradient (°) | 16.72 | 6.27 | 19.9 | 13.10 | 17.56 | 16.40 | 24.17 | 27.24 |
| Sand (%) | 44.87 | 39.44 | 38.22 | 55.33 | 46.42 | 46.19 | 52.66 | 39.96 |
| Silt (%) | 47.08 | 52.63 | 53.60 | 38.19 | 46.57 | 46.87 | 47.34 | 51.75 |
| Clay (%) | 8.06 | 7.93 | 8.18 | 6.49 | 7.01 | 6.95 | 7.40 | 8.30 |
| Organic (g/kg) | 7.04 | 5.31 | 5.75 | 6.30 | 13.30 | 8.91 | 5.99 | 8.10 |
| Soil bulk density (g/cm³) | 1.26 | 1.29 | 1.25 | 1.28 | 1.26 | 1.23 | 1.26 | 1.23 |
| Porosity | 0.48 | 0.46 | 0.48 | 0.47 | 0.49 | 0.48 | 0.49 | 0.49 |
| Mean canopy coverage (%) | 57.36 | 53.27 | 39.70 | 67.82 | 45.61 | 66.07 | 33.75 | 59.58 |
| Mean canopy height (m) | 0.59 | 1.83 | 3.58 | 0.68 | 1.73 | 1.85 | 3.02 | 11.77 |
| Mean tree DBH (cm) | - | - | 6.32 | - | - | - | 4.98 | 10.37 |
| Mean crown (cm) | - | - | 398.39 | - | 199.65 | 184.85 | 293.40 | 455.25 |
| Basal diameter (cm) | - | - | 10.17 | - | 1.31 | 3.76 | 8.13 | 12.85 |
| Planting density (/m²) | - | - | 30.5 | - | 129.67 | 262.40 | 36.17 | 58.66 |

[a] NG, FL, AO, PG, CK, SB, DP and BL refer to native grasses, farmland, apple
orchard, pasture grasses, *Caragana korshinskii*, sea buckthorn, David peach and
black locust, respectively.
**2.3 Data collection and analysis**
Each quadrat in the study area was covered by a single type of vegetation. Soil




moisture measurements in the growing season were made for the 5 m profile in 20
cm increments from July to August in 2014. Soil samples were sealed and taken to
the laboratory, and the gravimetric soil moisture content was determined using oven
drying at 105 ℃ to constant weight. Three sampling profiles were randomly chosen
to obtain the average soil moisture content for each sampling site. Meteorological
data (Fig. 2) were obtained during the sampling period by the MILOS520 weather
station located at the Ansai Research Station of Soil and Water Conservation
(109°19′23″E, 36°51′26″N). The average annual rainfall (2006-2013) was provided
by 29 rain gauges in or around the Ansai watershed, and the Ordinary Kriging
method was performed by ArcGIS10.0 to obtain the average annual rainfall at each
sampling site.

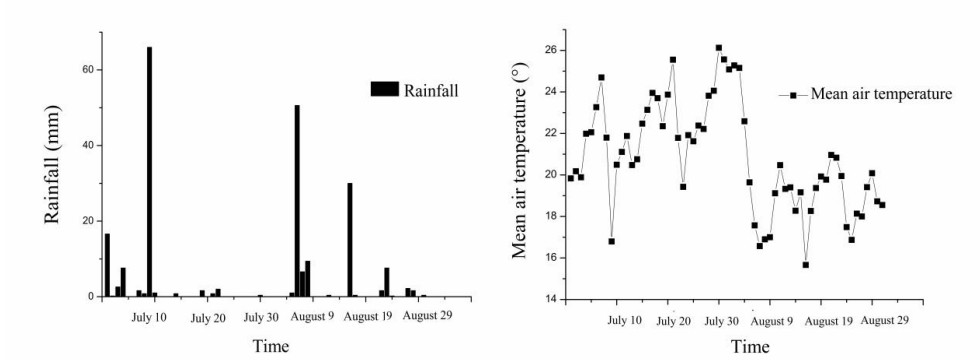

Figure 2. The rainfall (mm) and mean air temperature (℃) during the sampling
period.
Longitude, latitude and altitude were collected for each experimental site using
Garmin GPS (version eTrex 30). Slope gradients and slope aspects were determined
using the compass method in field investigation; slope gradients were transformed
into tan (slope), and slope aspects (clockwise from north) were transformed into cos
(aspect). At each sampling site, six undisturbed soil cores were collected from the
soil surface in metal cylinders (diameter 5 cm, length 5 cm) for measurements of
bulk density and porosity (Wang et al., 2008a). Bulk density and porosity were
determined from the volume–mass relationship for each core sample. Soil samples





were also collected at each sampling site. Soil particle size distributions were
measured using a laser scattering particle size distribution analyzer (BT-9300H,
Dandong, China). The proportions of clay (<0.002 mm), silt (0.002-0.02 mm), and
sand (>0.02 mm) content were then calculated. Soil organic matter content was
determined using the dichromate oxidation method (Hu et al., 2010). At each
sampling site, a vegetation investigation was also conducted. In forest sites, the stand
density (plants/ha), tree height (m), diameter at breast height (DHB, cm), basal
diameter (cm), under branch height (m), canopy width in a 20 m×20 m quadrat, and
total canopy or coverage of each quadrat were recorded. In shrub sites, the stand
density (plants/ha), plant height (m), basal diameter (cm), and canopy width in a 10
m×10 m quadrat were measured. Species composition, total herbaceous coverage,
grass height (m), litters and grass biomass were measured in each herbaceous
quadrat. The canopy cover was measured by visual estimation, and litter maximum
water holdup was measured using the immersion method.
**2.4 Statistical methods**
In this study, the depth-averaged soil moisture content ($SMC_d$) of each sampling
point was calculated using Eq. (1):

$$SMC_d = \frac{1}{k}\sum_{i=1}^{k} SMC_i,\qquad(1)$$

where $k$ is the number of measurement layers at site $j$, and $SMC_i$ is the mean soil
moisture content in layer $i$ calculated by using three random sampling profiles.
The depth-averaged soil moisture content for each vegetation type ($SMC_s$) was
calculated using Eq. (2):

$$SMC_s = \frac{1}{m}\sum_{j=1}^{m} SMC_{ij}\qquad(2)$$

where $m$ is the number of sampling points for each vegetation type (Table 1), and
$SMC_{ij}$ is the depth-averaged soil moisture content in layer $i$ at site $j$.
Soil moisture from each layer was pooled together for the 151 sampling





locations to conduct a descriptive analysis. Basic population statistics, such as minimum values (Min), maximum values (Max), mean values (Mean), standard deviations (SD), and coefficients of variation (CV), were reported for both the overall soil moisture datasets and those by vegetation type. SD and CV were employed to reflect the degree of spatial variability of soil moisture in different layers and different vegetation types (Ruan and Li, 2002). One-way ANOVA was used to assess the contribution of different vegetation cover types to the overall variation in soil moisture variables. Multiple comparisons were made using the least significant difference (LSD) method. To determine the contributing factors to soil moisture dynamics, spearman correlation analysis was first used to examine the relationships between soil moisture and environmental variables. Then, principle component analysis was performed to reduce the linear correlation that may exist among selected environment variables and to further identify a minimum data set (MDS) of environmental variables for each vegetation type. All statistical analyses were performed using SPSS (Version 20.0).

**3 Results**

**3.1 Summary statistics of soil moisture**

The summary statistics of soil moisture at various depths are given in Table 2. Kurtosis, skewness, and the Kolmogorov–Smirnov test value indicated that soil moisture data sets were normally distributed. Thus, statistical analysis could be performed without data transformation (Shi et al., 2014). In general, the mean soil moisture, SD, and CV were highly dependent on depth. The profile distributions of mean soil moisture content, SD, and CV are given in Table 2 and Fig. 3. The highest mean value (10.65%) was observed at the 20-40 cm depth, the lowest (8.15%) was at the 120-140 cm depth, and the mean soil moisture below 300 cm was almost constant. However, both SD and CV showed waving trends with increasing depth (Fig. 3). The profile distributions of SD and CV were consistent. The highest values of both occurred at 0-20 cm, 100-120 cm, and 480-500 cm (Table 2), which indicated that soil moisture at these depth ranges had relatively higher spatial





variability. Meanwhile, the lowest values occurred at 40-60 cm and 260-300 cm,
which indicated lower spatial variability of SMC at these depth ranges.
Table 2. Summary statistics of soil moisture at various depths in the Ansai
watershed.

| Depth (cm) | n [a] | Mean (%) | SD [b] (%) | Minimum (%) | Maximum (%) | CV [c] | K [d] | S | K-S |
|---|---|---|---|---|---|---|---|---|---|
| 0-20 | 151 | 9.78 | 3.87 | 2.76 | 20.73 | 0.40 | -0.32 | 0.35 | N(0.73) [e] |
| 20-40 | 151 | 10.65 | 2.91 | 3.68 | 18.98 | 0.27 | 0.03 | 0.06 | N(0.70) |
| 40-60 | 151 | 10.20 | 2.91 | 2.30 | 17.52 | 0.29 | -0.14 | -0.12 | N(0.59) |
| 60-80 | 151 | 9.35 | 3.25 | 2.97 | 17.53 | 0.35 | -0.50 | 0.04 | N(0.93) |
| 80-100 | 151 | 8.84 | 3.35 | 2.60 | 18.29 | 0.38 | -0.45 | 0.28 | N(0.95) |
| 100-120 | 151 | 8.21 | 3.31 | 3.29 | 18.23 | 0.40 | -0.27 | 0.57 | N(1.36) |
| 120-140 | 151 | 8.15 | 3.25 | 3.22 | 18.95 | 0.40 | 0.30 | 0.75 | N(0.93) |
| 140-160 | 151 | 8.16 | 3.09 | 3.37 | 18.56 | 0.38 | 0.40 | 0.80 | N(0.99) |
| 160-180 | 151 | 8.30 | 2.92 | 3.14 | 17.85 | 0.35 | 0.70 | 0.85 | N(1.06) |
| 180-200 | 151 | 8.47 | 2.70 | 3.22 | 17.89 | 0.32 | 1.48 | 1.01 | N(1.13) |
| 200-220 | 151 | 8.66 | 2.58 | 3.47 | 19.19 | 0.30 | 2.35 | 1.06 | N(1.23) |
| 220-240 | 151 | 8.83 | 2.54 | 3.59 | 19.72 | 0.29 | 2.99 | 1.05 | N(1.02) |
| 240-260 | 151 | 9.00 | 2.49 | 3.92 | 19.47 | 0.28 | 2.33 | 0.88 | N(0.94) |
| 260-280 | 151 | 9.00 | 2.37 | 4.08 | 18.46 | 0.26 | 1.94 | 0.74 | N(1.11) |
| 280-300 | 151 | 9.14 | 2.41 | 3.56 | 18.72 | 0.26 | 1.35 | 0.53 | N(0.65) |
| 300-320 | 151 | 9.15 | 2.46 | 3.26 | 18.08 | 0.27 | 1.45 | 0.54 | N(0.73) |
| 320-340 | 151 | 9.24 | 2.66 | 3.09 | 19.56 | 0.29 | 1.92 | 0.67 | N(0.81) |
| 340-360 | 151 | 9.36 | 2.83 | 2.98 | 19.38 | 0.30 | 1.31 | 0.59 | N(0.91) |
| 360-380 | 151 | 9.32 | 2.99 | 3.13 | 19.88 | 0.32 | 1.49 | 0.61 | N(0.91) |
| 380-400 | 151 | 9.35 | 3.09 | 2.81 | 20.85 | 0.33 | 1.99 | 0.60 | N(1.00) |
| 400-420 | 151 | 9.41 | 3.19 | 2.68 | 21.92 | 0.34 | 2.09 | 0.60 | N(0.80) |
| 420-440 | 151 | 9.33 | 3.21 | 2.70 | 20.97 | 0.34 | 1.43 | 0.55 | N(0.57) |
| 440-460 | 151 | 9.33 | 3.24 | 2.65 | 19.63 | 0.35 | 0.20 | 0.23 | N(0.73) |
| 460-480 | 151 | 9.35 | 3.43 | 2.67 | 19.88 | 0.37 | -0.08 | 0.26 | N(0.84) |
| 480-500 | 151 | 9.45 | 3.58 | 2.43 | 19.98 | 0.38 | -0.22 | 0.23 | N(0.87) |

Notes: [a] n refers to number of sampling points. [b] SD refers to standard deviation. [c]
CV refers to coefficient of variation. [d] K, S, K-S refer to Kurtosis, Skewness, and the
Kolmogorov-Smirnov test value, respectively. [e] N refers to normal distribution
(significance level is in parentheses).





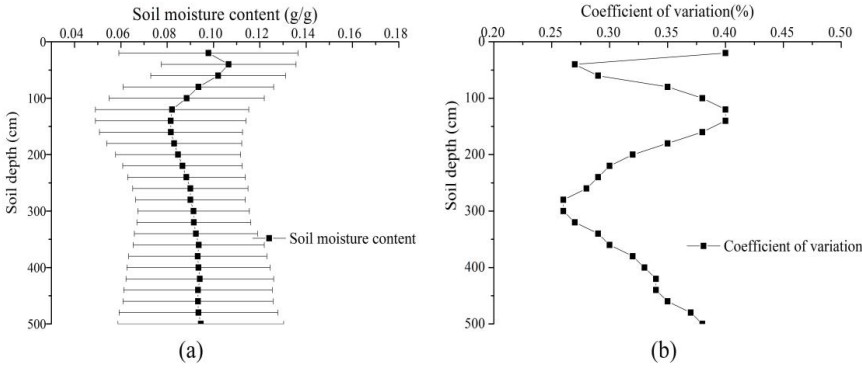

(a)                                                    (b)

Figure 3. The profile distribution of soil moisture content and coefficient of variation.

3                    Note: Error bar indicates standard deviation.

4         Moreover, different land management types greatly determined deep soil

moisture variation; the soil moisture statistics of various vegetation types under
different land management types are reported in Fig. 4. The results showed that the
depth-averaged SMC of native vegetation and human-managed vegetation were
significantly higher than that of introduced vegetation. In general, the mean soil
moisture of different vegetation covers was in the order:
FL>NG>AO>DP>SB>PG>BL>CK. The highest mean soil moisture existed in
farmland and the lowest in *Caragana korshinskii*. This result indicated that human
agricultural management measures can significantly improve soil moisture
conditions and that *Caragana korshinskii* was the most serious water consuming
species among the selected introduced vegetation types.





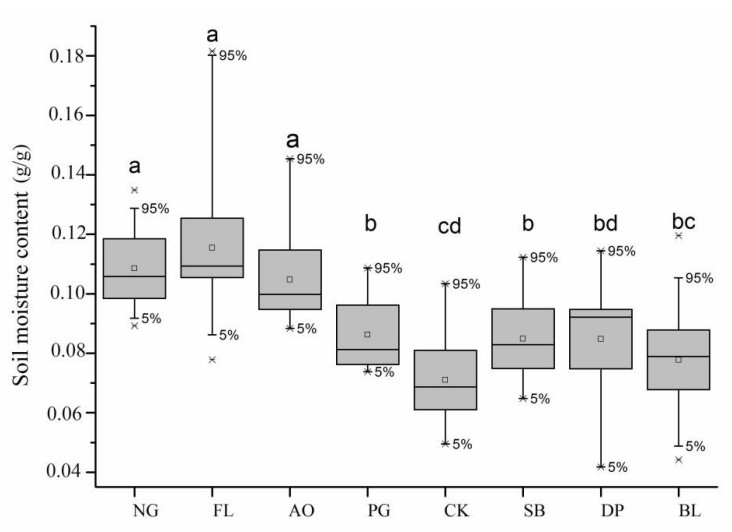

Figure 4. Soil moisture statistics for different vegetation types. Means with the same
letter above the box are not significantly different at the 0.05 significance level (LSD
test); NG, FL, AO, PG, CK, SB, DP and BL refer to native grasses, farmland, apple
orchard, pasture grasses, *Caragana korshinskii*, sea buckthorn, David peach and
black locust, respectively.
**3.2 Profile distribution of soil moisture by vegetation types**

8       According to a previous study, soil moisture profile characteristics are usually

complex in vegetation covering zones (Jia et al., 2013). Thus, soil moisture profiles
by vegetation types were chosen for analysis. As expected, the profile distribution
characteristics of deep soil moisture varied by vegetation type (Fig. 5).

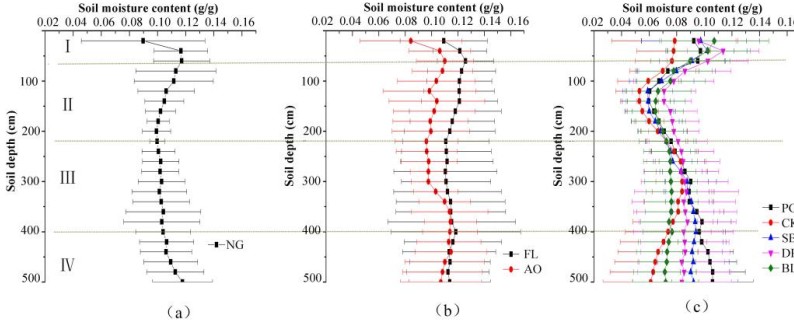





Figure 5. Profile distribution of mean soil moisture contents for different vegetation
types. Notes: (a) native grassland (NG-native grass), (b) human-managed vegetation
(FL-farmland, AO-apple orchard), (c) introduced vegetation (PG-pasture grass;
CK-*Caragana korshinskii*; SB-sea buckthorn; DP-David peach; BL-black locust).
Error bar indicates standard deviation, I-IV represent SMC at different soil layer
depth ranges (I: 0-60 cm, II: 60-120 cm, III: 120-400 cm, and IV: 400-500 cm), and
the dashed lines are the boundaries of different soil layer depth ranges.
Deep SMC in native grassland zones is seldom affected by vegetation due to
shallow root systems; thus, the deep SMC in native grasslands can be regarded as a
reference for local control (Yang et al., 2012b). Based on the inflection point of SMC
and the trending change of SD in native grasslands, the 5 m soil moisture profile was
divided into 4 layers, from I to V. (I) Shallow rapid change layer (0-60 cm); at this
layer, SMC increased as soil depth increased, while SD decreased as soil depth
increased. Moreover, this depth range is usually greatly influenced by rainfall events
and evaporation and is characterized as "rapid change" (Cantón et al., 2004; Entin et
al., 2000; Hébrard et al., 2006). (II) Main rainfall infiltration layer (60-220 cm); at
this layer, both SMC and SD decreased as soil depth increased, which indicated that
this layer may be a main rainfall infiltration layer. Furthermore, as depth increased,
the level of rainfall infiltration decreased. (III) Transition layer (220-400 cm); SMC
in this layer remained relatively constant as soil depth increased, but its SD increased
with soil depth, which indicated that this layer is unstable. We characterize it as a
transition layer. (IV) Stable layer (400-500 cm); this is a relatively stable layer whose
SD is constant as soil depth increases, despite increasing SMC with soil depth. At
this layer, SMC is seldom influenced by rainfall infiltration and evaporation. This
vertical stratification method of the soil moisture profile may not be ideal, but it can
reflect hydrological significance compared with previous studies (Yang et al., 2012a;
Yang et al., 2012b; Yang et al., 2014a).





The profile distribution characteristics of farmland were similar to those of
native grasslands, except for layer IV. Perhaps this is because management measures
increased the ranges of the rainfall infiltration layer. Similar profile distribution
characteristics were also found for apple orchards, except for the 300-500 cm layer.
As for vegetation-introduced, the profile distribution characteristics of the shallow
rapid change layer (0-60 cm) were more complex due to differences in evaporation
and rainfall redistribution caused by different vegetation coverage, while the deeper
layer (60-500 cm) could be generally divided into three categories: (1) as soil depth
increased, SMC decreased first and then increased (such as PG); (2) as soil depth
increased, SMC decreased first, then increased and finally became stable (such as SB,
DP, and BL); (3) as soil depth increased, SMC decreased first, then increased and
finally decreased again (such as CK). Different profile characteristics can reflect
different soil water consuming traits under different introduced vegetation.

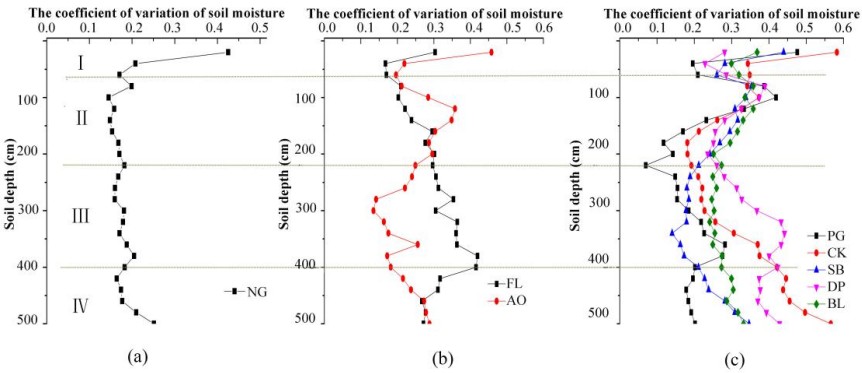

Figure 6. The coefficient of variation of soil moisture contents for different
vegetation types. Notes: (a) native grassland (NG-native grass), (b) human-managed
vegetation (FL-farmland, AO-apple orchard), (c) introduced vegetation (PG-pasture
grass; CK-*Caragana korshinskii*; SB-sea buckthorn; DP-David peach; BL-black
locust). I-IV represent SMC at different soil layer depth ranges (I: 0-60 cm, II: 60-120
cm, III: 120-400 cm, and IV: 400-500 cm); the dashed lines are the boundaries of
different soil layer depth ranges.



The spatial variation of SMC under different vegetation types displayed
different characteristics as well (Fig. 6). The spatial variation of native grassland was
clearly less than that in human-managed vegetation and introduced vegetation, and
the variation was relatively stable as depth increased, except for the shallow layer
(0-60 cm). In human-managed vegetation (farmland and orchard), the variation was
relatively higher and had a complex profile distribution due to different management
measures. However, the spatial variation in introduced vegetation was, to some
extent, consistent with the overall variation characteristics in this area (Fig. 5), which
indicates that introduced vegetation plays an important role in the spatial variation of
deep soil moisture in this area.
**3.3 Relationships between soil moisture content at different depth**
**ranges**
According to previous studies, shallow SMCs at different depths are usually
connected through infiltration and evapotranspiration processes (Shi et al., 2014).
However, the SMC relationships between the shallow layer and various deeper
layers have seldom been explored. Thus, the linear relationships of SMC at different
depths ranges (I-IV) were examined in the study area. The relationships between
point measurements at these depth ranges are shown in Fig. 7. Scatter plots suggest
that no correlations exist between the shallow layer (0-60 cm), moisture contents and
various deeper soil layer ranges ($R^2$ from 0.045 to 0.134). However, there were
positive and significant (P<0.01) correlations between moisture contents at different
soil depth ranges (60-220 cm, 220-400 cm and 400-500 cm). The correlations of the
neighboring layer ranges were relatively high, with $R^2$ from 0.68 to 0.78, while
much lower correlations of soil moisture values were observed between nonadjacent
depth ranges ($R^2 = 0.47$).





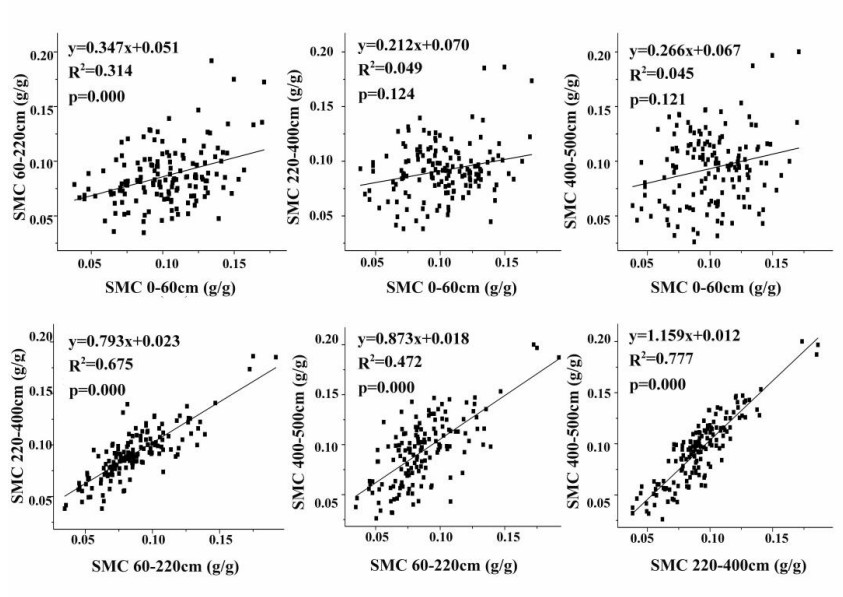

2 Figure 7. Correlations between point measurements at different depth ranges.

**3.4 Comparison of deep soil moisture content under different vegetation**
**types**

5  Generally, soil moisture at comparable soil depths was lower in introduced

vegetation (pasture grassland, shrub land and forestland) compared with native
grassland and human-managed vegetation (farmland and orchard). Farmland
(11.07-11.77%) had the highest SMC, followed by native grasses (10.47-11.19%).
The LSD-test indicated that soil moisture content in native grasses and farmland was
significantly higher than that in introduced vegetation ($P<0.05$, Table 3) at almost
every soil depth. Soil moisture varied from 7.56% to 10.4% in pasture grassland,
7.42-9.75% in sea buckthorn, 6.49-8.07% in *Caragana korshinskii*, 7.46-7.66% in
black locust, and 8.10-8.51% in David peach at layers of 60-500 cm. The LSD-test
indicated that there were significant differences in soil moisture at depths of 400-500
cm between different introduced vegetation types. For example, *Caragana*
*korshinskii* was significantly different from pasture grassland, sea buckthorn, and
David peach, while black locust was significantly different from pasture grassland
and sea buckthorn ($P<0.05$, Table 3).





Table 3. Soil moisture of 0-50 cm soil layers for different vegetation types.

| Land management types | Vegetation types | 0-60 cm | | | | 60-220 cm | | | | 220-400 cm | | | | 400-500 cm | | | |
|---|---|---|---|---|---|---|---|---|---|---|---|---|---|---|---|---|---|
| | | Min | Max | Mean | SD | Min | Max | Mean | SD | Min | Max | Mean | SD | Min | Max | Mean | SD |
| | | % | % | % | % | % | % | % | % | % | % | % | % | % | % | % | % |
| Native grasses | NG [a] | 6.74 | 16.95 | 11.15ab | 2.81 | 6.76 | 13.56 | 10.47a | 1.95 | 8.35 | 12.84 | 10.52ab | 1.62 | 8.17 | 14.72 | 11.19ab | 2.03 |
| Managed | FL | 9.27 | 17.10 | 11.9a | 2.19 | 6.91 | 19.19 | 11.07a | 3.28 | 7.78 | 18.62 | 11.07a | 3.20 | 7.53 | 20.01 | 11.77a | 3.58 |
| vegetation | AO | 5.84 | 13.09 | 10.01abc | 2.59 | 7.32 | 14.68 | 9.6ab | 2.40 | 7.72 | 14.06 | 10.45abc | 1.73 | 7.40 | 15.33 | 11.4ab | 2.26 |
| Introduced | PG | 6.35 | 12.80 | 9.43bcd | 2.15 | 6.81 | 8.36 | 7.56c | 0.52 | 7.69 | 13.14 | 8.97bcd | 1.55 | 8.49 | 14.29 | 10.4abc | 1.85 |
| vegetation | SB | 4.52 | 14.51 | 9.44cd | 2.77 | 5.15 | 10.74 | 7.42c | 1.64 | 7.11 | 12.09 | 8.93cd | 1.62 | 5.12 | 14.67 | 9.75bc | 2.64 |
| | CK | 3.82 | 13.15 | 7.9d | 2.84 | 5.05 | 10.50 | 7.25c | 1.35 | 4.94 | 11.62 | 8.07d | 2.11 | 2.63 | 12.50 | 6.49e | 2.92 |
| | BL | 4.81 | 15.31 | 10.21bc | 2.69 | 3.56 | 11.88 | 7.46c | 2.05 | 4.16 | 10.94 | 7.66d | 1.77 | 4.00 | 13.29 | 7.47de | 2.47 |
| | DP | 7.16 | 13.00 | 10.4abc | 2.04 | 3.47 | 10.80 | 8.1bc | 2.11 | 3.82 | 13.9`5 | 8.51d | 3.17 | 3.21 | 13.09 | 8.49cd | 3.24 |

Notes: [a] NG, FL, AO, PG, CK, SB, DP, and BL refer to native grasses, farmland,
apple orchard, pasture grasses, *Caragana korshinskii*, sea buckthorn, David peach
and black locust, respectively. Means with the same letter in the same column are not
significantly different at the 0.05 significance level (LSD).
As shown in Fig. 8, the SMC in farmland was higher than that in native
grassland, and soil desiccation occurred in all introduced vegetation. However, soil
desiccation varied among the vegetation types. In general, the soil moisture in layer
II (60-220 cm) was heavily consumed in almost all the introduced vegetation types.
PG and SB consumed less soil moisture in layers III-IV (220-500 cm) compared with
the three other introduced vegetation types, while the soil moisture in layers III-IV
(220-500 cm) of DP and BL consumed more consistently. Double layer soil
desiccation occurred in CK, indicating that the soil moisture in layers II and IV of CK
was heavily consumed, while the soil moisture in layer III was less consumed.
Furthermore, despite the deep root system of the apple orchard, soil desiccation did
not occur across the soil profile from 0-500 cm; even in the 320-450 cm layer, the
soil moisture in the apple orchard was higher than in native grasses.





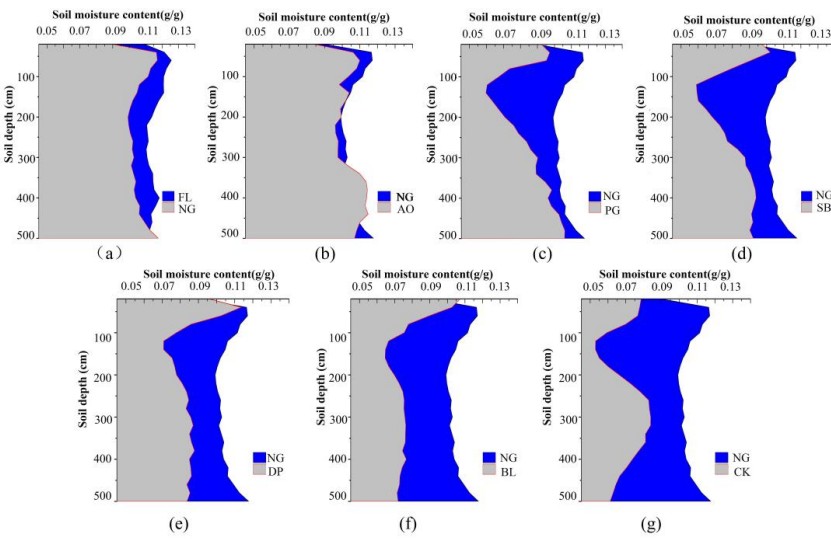

Figure 8. The comparison of soil moisture contents between human-managed
vegetation, introduced vegetation and native grasslands. Notes: (a) farmland (FL)
and native grasslands (NG), (b) apple orchard (AO) and native grasslands (NG), (c)
pasture grasslands (PG) and native grasslands (NG), (d) sea buckthorn (SB) and
native grasslands (NG), (e) David peach (DP) and native grasslands (NG), (f) black
locust (BL) and native grasslands (NG), (g) *Caragana korshinskii* (CK) and native
grasslands (NG).
**3.5 Spearman correlation coefficients between soil moisture and**
**selected environmental variables**

11       Spearman correlation coefficients were used to determine the strength of

possible relationships between soil moisture and selected variables. The correlation
analysis results are presented in Table 4, Table 5, and Table 6. The correlation
between soil moisture and environmental variations changed with soil depth and
vegetation type. In native grassland, the SMC in the shallow layer (0-60 cm)
showed significant correlations with average annual rainfall, while the SMC in the
deep layer showed significant correlations with altitude (60-500 cm), slope gradient
(220-500 cm), soil particle composition (60-500 cm), and average annual rainfall



(220-400 cm).
In farmland, the SMC in the shallow layer (0-60 cm) showed significant
correlations with altitude, clay content and bulk density, while the deep layers
(60-220 cm) were only influenced by bulk density. In areas of introduced vegetation,
apart from the significant correlations with topography, soil properties, and average
annual rainfall, the SMC showed different correlations with vegetation growth traits.
For instance, the SMC of BL showed significant negative correlations with plant
height (at 60-220 cm depth) and diameter at breast height (at 400-500 cm depth),
the SMC of DP showed significant negative correlations with crown width (at 0-60
cm depth) and basal diameter (at 0-60 cm depth), and the SMC of SB showed a
significant negative correlation with plant density (at 60-500 cm depth). A
significant correlation was found between aspect and SMC in some introduced
vegetation in PG (at 400-500 cm depth) and BL (at 60-400 cm depth). Moreover,
positive correlations existed between SMC and soil surface conditions; for instance,
SMC of DP showed significant correlations with grass biomass (at 400-500 cm
depth), SMC of AO showed significant correlations with litter biomass (at 400-500
cm depth), and SMC of CK showed significant correlations with litter max water
holding (at 220-500 cm depth). Furthermore, in apple orchards, both soil buck
density (at 60-400 cm depth) and porosity (at 60-220 cm depth) showed significant
correlations with SMC.
Table 4. Spearman correlation coefficients between soil moisture (grassland,
farmland and pasture grassland) and selected environmental variables.

| | Native grasslands | | | | Farmland | | | | Pasture grassland | | | |
|---|---|---|---|---|---|---|---|---|---|---|---|---|
| | I | II | III | IV | I | II | III | IV | I | II | III | IV |
| Altitude | 0.27 | -0.49 | -0.56 | -0.53 | -0.51 | -0.37 | -0.30 | -0.19 | -0.04 | -0.19 | -0.06 | 0.08 |
| Slope position | 0.37 | 0.11 | -0.11 | -0.07 | 0.14 | 0.20 | 0.28 | 0.41 | 0.21 | -0.14 | -0.32 | 0.02 |
| Cos (Aspect) | 0.03 | -0.22 | -0.35 | -0.44 | -0.27 | 0.06 | 0.03 | 0.21 | 0.14 | 0.07 | 0.64 | 0.86 |
| Tan (Slope) | 0.04 | 0.36 | 0.67 | 0.59 | 0.09 | -0.21 | -0.07 | 0.21 | 0.02 | -0.37 | 0.09 | 0.34 |
| Clay | 0.09 | 0.67 | 0.56 | 0.43 | 0.43 | 0.33 | 0.37 | 0.22 | 0.33 | 0.13 | 0.54 | 0.46 |
| Silt | 0.07 | 0.56 | 0.37 | 0.27 | 0.13 | 0.24 | 0.38 | 0.38 | 0.17 | 0.13 | 0.66 | 0.59 |
| Sand | -0.09 | -0.62 | -0.42 | -0.32 | -0.17 | -0.24 | -0.35 | -0.35 | -0.25 | -0.13 | -0.58 | -0.47 |
| Organic | 0.02 | -0.18 | -0.30 | -0.19 | 0.08 | 0.08 | -0.13 | -0.23 | -0.36 | -0.04 | -0.28 | -0.64 |
| Soil bulk density | -0.11 | -0.06 | -0.07 | -0.04 | 0.49 | 0.45 | 0.31 | 0.34 | -0.16 | 0.14 | -0.01 | -0.16 |




| | | | | | | | | | | | | |
|---|---|---|---|---|---|---|---|---|---|---|---|---|
| Porosity | 0.10 | 0.07 | 0.06 | 0.05 | -0.35 | -0.33 | -0.26 | -0.20 | -0.08 | -0.53 | -0.26 | -0.12 |
| Annual average rainfall | -0.43 | -0.01 | 0.46 | 0.37 | 0.20 | -0.05 | -0.11 | -0.23 | 0.01 | -0.47 | 0.15 | 0.36 |
| Vegetation coverage | 0.01 | -0.19 | -0.08 | -0.02 | 0.39 | 0.15 | 0.11 | 0.26 | -0.57 | -0.38 | 0.37 | 0.11 |
| Grass biomass | -0.38 | -0.07 | 0.20 | 0.08 | 0.22 | -0.05 | -0.06 | -0.06 | -0.49 | -0.01 | 0.28 | -0.10 |
| Grass height | 0.35 | 0.33 | 0.01 | 0.00 | 0.20 | 0.04 | 0.06 | 0.15 | -0.06 | -0.23 | 0.46 | 0.32 |

Notes: I, II, III, and IV represent SMC at different soil layer depths; among them, I
means SMC at 0-60 cm, II means SMC at 60-220 cm, III means SMC at 220-400 cm,
and IV means SMC at 400-500 cm. Significant correlations (P<0.05) are shown in
bold, and significant correlations (P<0.01) are shown in bold with underline.
Table 5. Spearman correlation coefficients between soil moisture (shrub land) and
selected environmental variables.

| | *Caragana korshinskii Kom* | | | | Sea buckthorn | | | |
|---|---|---|---|---|---|---|---|---|
| | I | II | III | IV | I | II | III | IV |
| Altitude | 0.06 | -0.34 | **_-0.70_** | **-0.59** | -0.15 | **_-0.68_** | **-0.56** | -0.33 |
| Slope position | 0.34 | 0.27 | -0.08 | -0.11 | 0.10 | -0.15 | -0.25 | -0.35 |
| Cos (Aspect) | 0.11 | 0.38 | 0.34 | 0.32 | 0.43 | 0.29 | 0.34 | 0.07 |
| Tan (Slope) | -0.11 | 0.06 | -0.10 | -0.05 | -0.06 | -0.44 | -0.19 | 0.00 |
| Clay | 0.23 | 0.09 | -0.24 | -0.09 | **0.55** | 0.24 | 0.22 | -0.02 |
| Silt | -0.04 | 0.14 | 0.32 | **0.53** | 0.29 | **0.56** | **0.51** | 0.41 |
| Sand | -0.02 | -0.14 | -0.23 | -0.45 | -0.31 | **-0.56** | -0.48 | -0.37 |
| Organic | -0.29 | 0.07 | **0.47** | **0.49** | 0.13 | 0.24 | 0.28 | -0.20 |
| Soil bulk density | 0.22 | 0.12 | -0.23 | -0.24 | 0.01 | 0.06 | -0.28 | -0.18 |
| Porosity | -0.04 | -0.01 | 0.13 | 0.14 | 0.00 | -0.07 | 0.20 | 0.02 |
| Annual average rainfall | 0.36 | **0.56** | 0.23 | 0.19 | -0.28 | 0.16 | 0.22 | 0.18 |
| Litter biomass | -0.23 | -0.17 | -0.04 | 0.10 | 0.44 | -0.28 | -0.33 | -0.39 |
| Litter max water holding | -0.02 | 0.31 | **0.59** | **_0.60_** | -0.21 | -0.13 | 0.09 | 0.08 |
| Vegetation coverage | -0.07 | -0.03 | 0.06 | -0.03 | 0.15 | -0.02 | -0.14 | -0.16 |
| Grass biomass | 0.03 | 0.20 | 0.42 | 0.45 | -0.01 | 0.45 | 0.26 | 0.31 |
| Grass height | 0.01 | 0.22 | 0.35 | 0.43 | -0.29 | 0.11 | 0.06 | 0.18 |
| Plant height | 0.03 | 0.26 | 0.24 | 0.23 | -0.05 | -0.02 | 0.25 | 0.09 |
| Crown width | 0.02 | 0.21 | 0.24 | 0.30 | -0.48 | -0.29 | 0.12 | 0.07 |
| Basal diameter | **-0.49** | -0.23 | 0.31 | 0.40 | -0.49 | -0.28 | 0.06 | -0.01 |
| Plant density | -0.18 | -0.28 | 0.08 | -0.09 | -0.31 | **_-0.69_** | **-0.57** | **-0.56** |

Notes: I, II, III, and IV represent SMC at different soil layer depths; among them, I
means SMC at 0-60 cm, II means SMC at 60-220 cm, III means SMC at 220-400 cm,
and IV means SMC at 400-500 cm. Significant correlations (P<0.05) are shown in





bold, and significant correlations (P<0.01) are shown in bold with underline.
Table 6. Spearman correlation coefficients between soil moisture (orchard land and
forest) and selected environmental variables.

| | Apple orchard | | | | Black locust | | | | David peach | | | |
|---|---|---|---|---|---|---|---|---|---|---|---|---|
| | I | II | III | IV | I | II | III | IV | I | II | III | IV |
| Altitude | 0.14 | -0.62 | -0.25 | -0.16 | -0.12 | -0.07 | -0.07 | 0.20 | 0.43 | -0.14 | 0.05 | 0.06 |
| Slope position | -0.36 | 0.11 | 0.34 | 0.14 | 0.16 | -0.20 | -0.22 | -0.21 | -0.56 | -0.34 | -0.50 | -0.55 |
| Cos (Aspect) | 0.38 | 0.02 | -0.01 | 0.35 | 0.05 | **0.34** | **0.34** | 0.22 | 0.22 | 0.07 | 0.13 | 0.30 |
| Tan (Slope) | **-0.77** | -0.28 | 0.26 | 0.33 | -0.17 | -0.07 | -0.17 | **-0.41** | -0.31 | -0.15 | 0.19 | 0.07 |
| Clay | 0.50 | **0.87** | 0.42 | -0.25 | 0.19 | 0.23 | 0.13 | -0.09 | **0.76** | 0.30 | 0.15 | 0.06 |
| Silt | 0.16 | **0.81** | **0.67** | 0.08 | 0.25 | 0.27 | 0.14 | -0.15 | **0.69** | 0.44 | 0.42 | 0.27 |
| Sand | -0.16 | **-0.81** | **-0.67** | -0.08 | -0.25 | -0.24 | -0.14 | 0.13 | **-0.69** | -0.44 | -0.42 | -0.27 |
| Organic | 0.31 | **0.66** | 0.38 | 0.13 | -0.29 | 0.00 | 0.02 | -0.22 | 0.48 | -0.04 | -0.12 | -0.35 |
| Soil bulk density | -0.27 | **-0.65** | **-0.82** | -0.32 | 0.20 | -0.27 | -0.08 | -0.06 | -0.48 | -0.25 | -0.43 | -0.41 |
| Porosity | 0.41 | **0.86** | 0.49 | -0.06 | -0.19 | 0.29 | 0.14 | 0.00 | 0.48 | 0.38 | 0.52 | 0.30 |
| Annual average rainfall | 0.08 | 0.29 | -0.07 | -0.38 | -0.04 | -0.02 | 0.26 | -0.12 | 0.24 | 0.17 | -0.11 | -0.42 |
| Litter biomass | -0.45 | 0.23 | 0.47 | **0.72** | 0.23 | -0.05 | 0.13 | -0.03 | -0.06 | 0.01 | 0.08 | 0.08 |
| Litter max water holding | 0.18 | 0.32 | 0.08 | 0.33 | -0.22 | 0.28 | 0.21 | 0.20 | **0.59** | 0.14 | 0.13 | 0.27 |
| Vegetation coverage | -0.53 | -0.54 | 0.10 | -0.01 | -0.03 | -0.12 | 0.11 | -0.03 | -0.38 | -0.39 | -0.47 | -0.41 |
| Grass biomass | **-0.66** | -0.22 | 0.03 | 0.39 | 0.12 | 0.03 | 0.07 | 0.30 | 0.15 | 0.46 | **0.80** | 0.55 |
| Grass height | 0.18 | -0.03 | -0.17 | -0.62 | 0.28 | -0.01 | 0.02 | 0.08 | -0.02 | -0.50 | -0.01 | -0.01 |
| Plant height | 0.41 | 0.26 | -0.09 | -0.49 | -0.29 | **-0.35** | 0.11 | 0.05 | -0.56 | -0.34 | -0.11 | -0.01 |
| Diameter at breast height | 0.16 | 0.62 | 0.31 | 0.04 | -0.20 | -0.29 | -0.03 | **-0.34** | -0.57 | -0.36 | -0.24 | -0.15 |
| Crown width | 0.03 | 0.49 | 0.29 | 0.15 | -0.10 | -0.26 | 0.07 | -0.07 | **-0.59** | -0.50 | -0.36 | -0.29 |
| Basal diameter | 0.13 | 0.54 | 0.22 | 0.07 | -0.17 | -0.23 | 0.03 | -0.25 | **-0.61** | -0.42 | -0.20 | -0.07 |
| Plant density | -0.35 | -0.56 | -0.20 | -0.15 | 0.15 | 0.05 | 0.03 | 0.18 | 0.09 | 0.07 | 0.05 | -0.08 |

Notes: I, I, III, and IV represent SMC at different soil layer depths; among them, I
means SMC at 0-60 cm, II means SMC at 60-220 cm, III means SMC at 220-400 cm,
and IV means SMC at 400-500 cm. Significant correlations (P<0.05) are shown in
bold, and significant correlations (P<0.01) are shown in bold with underline.
**3.6 Principal component analysis (PCA)**
Based on spearman correlation analysis, only environmental variables that
showed significant correlations (P<0.05) with SMC were retained for further
analysis. There were 9 environmental variables for grassland and farmland (Group 1),
9 environmental variables for shrub land (Group 2), and 15 environmental variables



for forestland and orchard (Group 3). Among these variables, some were linearly
correlated. Thus, the dimensionality of these data sets could be reduced. Following
Hu et al. (2010) and Xu et al. (2008), principal component analysis was performed to
obtain a MDS of environmental variables; the results are listed in Table 7. Note that
only principal components (PCs) with eigenvalues N>1.0 and only variables with
highly weighted factor loading (i.e., those with absolute values for factor loading
within 10% of the highest value) were retained for the MDS (Mandal et al., 2008;
Shi et al., 2014). For Group 1, the PCA identified four PC that accounted for 80.04%
of the variance, of which the first three PCs accounted for most of this variance
(68.32%); for Group 2, four PCs, accounting for 84.39% of the variance, were
identified; for Group 3, five PCs, accounting for 74.54% of the variance, were
identified. In grassland and farmland, PC#1 included 3 variables that had highly
weighted factor loadings, including clay, silt, and sand, which indicates that soil
particle composition was the most important factor influencing soil moisture
variation. Under PC#2, PC#3, and PC#4, only one variable for each principal
component had a high factor loading: slope aspect, annual average rainfall, and soil
buck density, respectively. In shrub land, the highly weighted factor loadings of
PC#1 were clay, silt, and sand, while altitude and plant density were the highly
weighted factor loadings for PC#2. Under PC#3 and PC#4, only one variable from
each had a high factor loading: litter max water holding and organic, respectively. In
forest and orchard land, diameter at breast height, basal diameter, and sand content
accounted for the highly weighted factor loadings of PC#1; porosity was the only
variation that accounted for the highly weighted factor loadings of PC#2. As for
PC#3, there were four variations that were highly weighted: clay, silt, soil buck
density, and litter max water holding. Under PC#4 and PC#5, only one variable from
each had a high factor loading: slope aspect and slope gradient, respectively.



Table 7. Principle component analysis (PCA) of environmental attributes

| Principal component | Group 1: grassland and farmland | | | | Group 2: shrub land | | | | Group 2: orchard, and forest | | | | |
|---|---|---|---|---|---|---|---|---|---|---|---|---|---|
|  | PC[a] #1 | PC #2 | PC #3 | PC #4 | PC #1 | PC #2 | PC #3 | PC #4 | PC #1 | PC #2 | PC #3 | PC #4 | PC #5 |
| Eigenvalue | 3.58 | 1.45 | 1.13 | 1.05 | 2.99 | 2.32 | 1.27 | 1.01 | 4.51 | 2.62 | 1.80 | 1.17 | 1.08 |
| % of variance | 39.75 | 16.07 | 12.50 | 11.71 | 33.25 | 25.74 | 14.16 | 11.25 | 30.09 | 17.49 | 11.97 | 7.78 | 7.23 |
| Cumulative % | 39.75 | 55.82 | 68.32 | 80.04 | 33.25 | 58.98 | 73.14 | 84.39 | 30.09 | 47.57 | 59.54 | 67.32 | 74.54 |
| Factor loading/eigenvector | | | | | | | | | | | | | |
| Annual average rainfall | 0.21 | -0.50 | **0.71** | 0.06 | 0.23 | -0.52 | 0.41 | -0.56 | | | | | |
| Altitude | -0.46 | 0.23 | 0.53 | 0.51 | -0.08 | **0.83** | -0.17 | -0.05 | | | | | |
| Slope aspect | -0.16 | **0.81** | 0.24 | -0.07 | | | | | -0.06 | -0.05 | 0.21 | **0.64** | -0.10 |
| Slope gradient | 0.64 | -0.43 | -0.04 | -0.23 | | | | | 0.55 | 0.22 | 0.07 | 0.01 | **-0.70** |
| Clay | **0.86** | 0.25 | -0.13 | 0.15 | **0.95** | 0.11 | 0.01 | 0.10 | 0.64 | -0.44 | **-0.50** | -0.03 | -0.04 |
| Silt | **0.93** | 0.27 | 0.05 | 0.09 | **0.97** | 0.05 | 0.12 | 0.07 | 0.68 | -0.44 | **-0.45** | 0.28 | 0.08 |
| Sand | **-0.94** | -0.27 | -0.02 | -0.11 | **-0.98** | -0.06 | -0.11 | -0.08 | **-0.77** | 0.39 | 0.37 | -0.23 | -0.17 |
| Organic | -0.48 | -0.03 | -0.50 | 0.49 | -0.15 | -0.42 | 0.26 | **0.80** | 0.54 | -0.23 | 0.28 | -0.16 | -0.12 |
| Soil bulk density | -0.41 | 0.31 | 0.07 | **-0.67** | | | | | -0.25 | 0.61 | **-0.49** | 0.29 | -0.05 |
| Porosity | | | | | | | | | 0.38 | **-0.74** | 0.22 | -0.37 | -0.13 |
| Litter biomass | | | | | | | | | 0.54 | 0.43 | -0.29 | -0.04 | 0.20 |
| Litter max water holding | | | | | -0.28 | -0.13 | **0.81** | 0.00 | -0.24 | -0.48 | **0.46** | 0.31 | 0.50 |
| Grass biomass | | | | | | | | | 0.27 | 0.49 | -0.17 | -0.37 | 0.39 |
| Plant height | | | | | | | | | 0.69 | 0.31 | 0.34 | -0.22 | 0.24 |
| Diameter at breast height | | | | | | | | | **0.80** | 0.34 | 0.31 | 0.14 | 0.05 |
| Crown width | | | | | | | | | 0.39 | 0.20 | 0.43 | 0.22 | -0.09 |
| Basal diameter | | | | | -0.13 | 0.74 | 0.53 | -0.07 | **0.75** | 0.37 | 0.26 | 0.18 | 0.07 |
| Plant density | | | | | -0.04 | **0.77** | 0.21 | 0.15 | | | | | |

Notes: [a] PC refers to principal component. Significant correlations (P<0.05) are
shown in italics, and significant correlations (P<0.01) are shown in bold. Factor
loadings in bold are considered highly weighted when within 10% of variation of the
absolute values of the highest factor loading in each PC.

6         In total, 6 out of 9 environmental variables for grassland and farmland (group 1),

7 out 9 for shrub (group 2), and 10 out of 15 for forest and apple orchard (group 3)
were selected as MDS variables. Moreover, the MDS variables for each vegetation
type were selected (Table 8). It can be concluded that, at the watershed scale, the
main influencing factors of SMC variation under native grasslands were soil particle
composition (clay, silt, and clay content) and average annual rainfall. In farmland,
the dominant influencing factors were clay content and soil buck density. For
introduced vegetation types, the main influencing factors were more complex; apart
from soil texture and physical characteristics, topographical factors and vegetation





traits also strongly affected SMC variation. Moreover, the main influencing depth
ranges of different environmental factors varied with vegetation types (Fig. 9). For
example, in native grasslands and apple orchard land, soil particle size composition
mainly influenced deep SMC at 60-220 cm, while in pasture grassland, the most
significant influencing depths were 220-400 cm. This indicates that vegetation
coverage or human management measures can alter the depths of environmental
factors influencing SMC.
Table 8. The minimum data set of environmental variables.

| Vegetation types | Influencing variables |
|---|---|
| Native grasses | Cl[a], Sl, Sa, AAR |
| Farmland | Cl, SBD |
| Apple orchard | SG, Cl, Sl, Sa, SBD, Po |
| Pasture grasses | SA, Sl |
| Sea buckthorn | Al, Sl, Sa, Cl, PD |
| *Caragana korshinskii* | Al, Sl, Or, LMWH |
| Black locust | SA, SG, DBH |
| David peach | Cl, Sa, BD, LMWH |

Note: [a] Cl, SA, SG, Sl, Sa, Or, Po, SBD, DBH, BD, LMWH refer to clay, slope
aspect, slope gradient, silt, sand, organic, porosity, soil bulk density, diameter at
breast height, basal diameter, and litter max water holding, respectively.





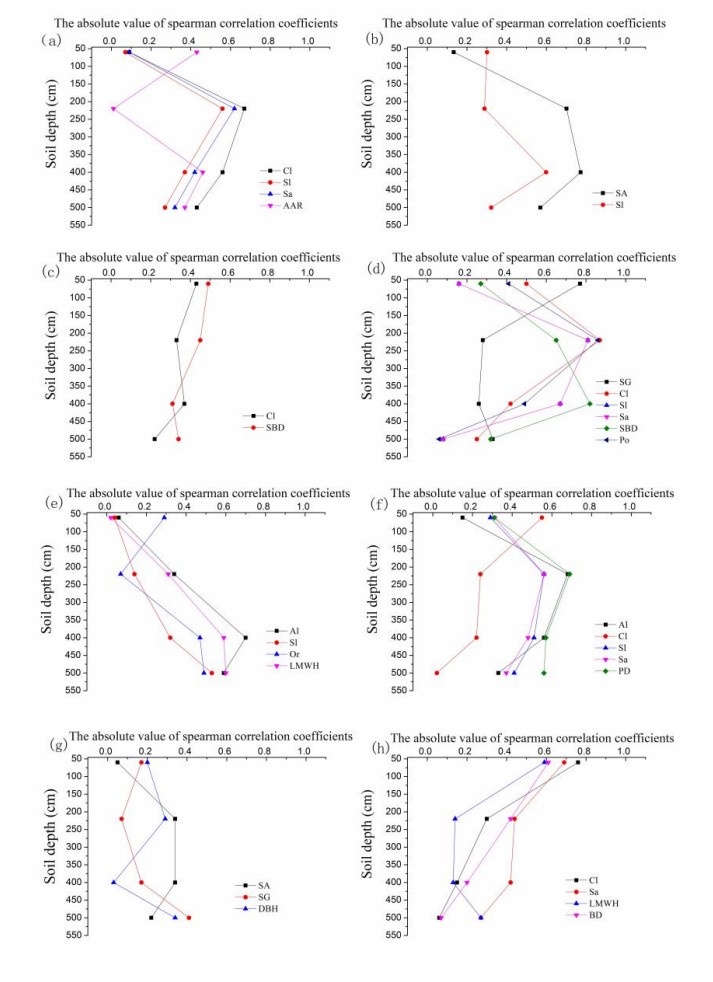

Figure 9. The influencing depths of the minimum data set of environmental variables
for soil moisture content of different vegetation types. Notes: (a) Native grasslands:
Cl-clay, Sl-silt, Sa-sand, AAR-annual average rainfall. (b) Pasture grasses: Sa-sand,
Sl-silt. (c) Farmland: Cl-clay, SBD-soil bulk density. (d) Apple orchard: SG-slope
gradient, Cl-clay, Sl-silt, Sa-sand, SBD-soil bulk density, Po-porosity. (e) *Caragana*
*korshinskii*: Al-altitude, Sl-silt, Or-organic, LMWH-litter max water holding. (f) Sea
buckthorn: Al-altitude, Cl-clay, Sl-silt, Sa-sand, PD-plant density. (g) Black locust:
SA-slope aspect, SG-slope gradient, DBH-diameter at breast height. (h) David peach:
Cl-clay, Sa-sand, BD-basal diameter, LMWH-litter max water holding.





## 4 Discussion

### 4.1 Spatial variation characteristics of deep soil moisture at the watershed scale

The spatial variation of deep soil moisture at the watershed scale varied with soil depth (Fig. 3 and Fig. 5). The shallow layer (0-60 cm) is more susceptible to soil evaporation or rainfall, and rainfall at this layer can be evapotranspired rapidly; thus, SMC at the surface layer increased as soil depth increased. At a soil depth of 60-220 cm, the influence of soil evaporation was relatively weak, rainfall infiltration could be stored in soil without the strong consumption of vegetation, and rainfall infiltration decreased as soil depth increased. Meanwhile, the soil layer at 220-400 cm was less influenced by rainfall infiltration; thus, SMC remained constant as soil depth increased. Soil depth below 400 cm was a deep stable SMC storage layer (Fig. 5). Moreover, compared with the rapid change of SMC caused by rainfall infiltration and evapotranspiration in the shallow layer, rainfall infiltration and evapotranspiration were usually slow processes in the deeper soil layers. This hysteresis process in deeper soil layers decreased the correlation relationship of SMC between the shallow and deeper layers (Fig. 7). However, the existence of deep rooted vegetation and human agricultural management measures altered the vertical SMC distribution rules, resulting in more complex spatial variation (Fig. 3). The highest variation of SMC at this watershed occurred at 0-20 cm, 100-120 cm, and 480-500 cm. Surface SMC (0-20 cm) was more prone to daily soil evaporation and rainfall events; different sampling climates and vegetation cover conditions contributed to the high variation (Cantón et al., 2004; Entin et al., 2000; Hébrard et al., 2006). However, the SMC was the lowest in the 120-140 cm layer, with high variation. This result is inconsistent with previous studies, which reported that high spatial variations usually appear in higher SMC and decrease when SMC becomes lower (Ibrahim and Huggins, 2011). This is likely because the most serious soil desiccation occurred in this layer for all introduced vegetation types (Fig. 5 and Fig. 8), increasing their difference with native grasses and human management vegetation



types, eventually resulting in high variation. While the high variation at 400-500 cm
may have been mainly caused by the different water consuming capacities of
different vegetation types, this depth range is rarely influenced by rainfall event
infiltration and soil evaporation (Chen et al., 2008a; Wang et al., 2009).

5       Soil moisture content spatial variation traits varied with vegetation types as well.

In native grassland, only the surface layer displayed high soil moisture spatial
variations (Fig. 5.), while the soil moisture variation at the deep soil layer was
relatively low and stable. Usually, the roots of native grasses are distributed at 0-50
cm (Han et al., 2009). Thus, soil moisture below this depth is seldom influenced by
vegetation transpiration, and local control, such as topography factors, soil factors,
and climate conditions, may contribute to the spatial variation of SMC. In farmland,
the SMC and its spatial variation were higher than that in native grassland, indicating
that human agricultural measures can greatly increase SMC and its spatial variation.
For introduced vegetation, the SMC was significantly lower than that in native
grassland, indicating that soil desiccation occurred for all introduced vegetation.
Moreover, different introduced vegetation showed different soil desiccation traits
(Fig. 8). This result is different from that of a previous study (Yang et al., 2012b),
which reported that no significant differences existed among different introduced
vegetation. This was probably caused by the difference in annual precipitation; the
mean annual precipitation of Yang's study area was 386 mm, which is far less than in
our study area (505 mm). The lower annual precipitation resulted in plants not
getting enough water, eventually leading to more homogeneous soil desiccation
among the different introduced vegetation. Among the selected introduced
vegetation types, *Caragana korshinskii* consumed the most water (Fig. 4.); this
partly disagrees with most previous studies (Wang et al., 2010b; Wang et al., 2011b;
Wang et al., 2009; Yang et al., 2012b), which reported that forest consumes more soil
moisture than shrub land. This discrepancy may have been due to the higher planting
density of CK in our study area. Moreover, the main difference in soil desiccation
under introduced vegetation occurred at 60-220 cm and 400-500 cm (Table 3, Fig.



8.); this contributed to the higher spatial variation of SMC at these two layers (Fig.

2    3).

**4.2 Mechanisms of deep soil moisture spatial variability**

The spatial variation of deep SMC is the combined result of topography factors,
soil factors, vegetation factors, and climate conditions. In this study, vegetation
coverage was an important factor influencing deep soil moisture variation. The effect
of vegetation on soil moisture is shown in many aspects. First, due to the existence
of a root system, soil moisture consumption in vegetation coverage zones is usually
higher than that in zones lacking vegetation coverage (Savva et al., 2013), and
different root systems determine different soil moisture consumption traits for
various vegetation types (Fig. 8). For example, the roots of native grasses are usually
distributed from 0-50 cm (Han et al., 2009), those of farmland from 0-40 cm (Feng
et al., 2007), and those of Alfalfa and *Caragana korshinskii* can reach 3 m and 6 m,
respectively (Wang et al., 2010b; Yang et al., 2014b). Thus, introduced vegetation
with a deep root system consumes more and deeper soil moisture than farmland and
native grasses (Table 3). Individual vegetation growth conditions and planting
density can also influence deep SMC variation. For example, deep soil moisture in
BL showed negative correlations with plant height and diameter at breast height,
while SB showed negative correlations with plant density (Table 5 and Table 6). This
phenomenon indicates that, in the deeper root system, forest individual growth
conditions mainly explain SMC consumption, while planting density mainly
accounts for SMC consumption in the less deep root system of shrubs. In addition to
SMC consumption, the canopy interception system and surface coverage system can
also have positive influences on soil moisture (Mart nez-Fern ndez and Ceballos,
2003; Starks et al., 2006). Generally, a well-developed surface O-layer can hold
more soil moisture; thick litter, humus layer and forest grass can retain more rainfall
for infiltration as well as reduce soil evaporation (Vivoni et al., 2008). In this study,
litter biomass, water holding capacity, and forest grass all showed different degrees
of significant positive correlations with deep soil moisture for different vegetation





types (Table 4, Table 5, and Table 6).

2        Climate factors that affect soil moisture are mainly determined by differences in

rainfall infiltration and solar radiation (Savva et al., 2013). According to previous
studies, deep soil moisture is relatively stable compared with the shallow layer,
especially at depths below 200 cm. For example, Chen et al. (2008a) found that
rainfall only affects the depth of 0-200 cm during drought years. Based on six years
of observation in this region (Wang et al., 2009), it was also found that no significant
changes occur in soil moisture below 200 cm. Thus, soil moisture in deeper layers is
seldom influenced by rainfall events. However, in this study, SMC in the deep soil
layer (60-220 cm in *Caragana korshinskii* and 220-400 cm in native grasslands)
showed significant positive correlations with the six-year average annual rainfall,
which indicates that deep SMC may be a long-term result of a water budget surplus.

13       Topography is another important factor that greatly affects the redistribution and

consumption of soil moisture (Qiu et al., 2001; Zhu et al., 2014b). Slope position,
altitude, and slope gradient mainly affect the lateral flow of soil moisture. Lower
position or latitude usually has a higher soil moisture content (He et al., 2003; Zhu et
al., 2014a), while slope gradient usually shows a negative correlation with SMC,
indicating that a steep slope usually has a lower SMC than a gentle slope (Kim et al.,
2007). As for the slope aspect, different aspects are usually caused by changes in
solar radiation (Yang et al., 2012c), resulting in different rates of soil moisture
evaporation. Thus, SMC on a sunny slope is usually lower than on a shady slope
(Galicia et al., 1999; Wang et al., 2008a; Zhao et al., 2007). In this study, altitude had
negative correlations with deep soil moisture; slope gradient showed significant
positive correlations with SMC in grasslands (220 cm-500 cm), while significant
negative correlations were found in black locust (400-500 cm). This indicates that
the introduced vegetation can alter the topography factors' influence on SMC
variation; this was also verified by Yang et al. (2012c), who found that introduced
vegetation can lead to homogeneity of the deep SMC. This was true for slope aspect,
which only showed positive correlations with SMC in pasture grasses (400-500 cm)





and black locust (60-400 cm). Moreover, different soil traits determine different
water transmission and conservancy characteristics, which may greatly influence the
flow or storage of water in soil (Western et al., 2004). For example, Gómez-Plaza et
al. (2001) found that soil porosity has a significant relationship with soil moisture in
wet areas. Meanwhile, Vachaud et al. (1985) found that soil texture, especially clay
content, is an important influencing factor of soil moisture variation. It was also
found that soil layers with higher clay content usually have higher soil moisture
(Ojha et al., 2014). In this study, soil particle composition was an important
influencing factor of deep SMC variation at the watershed scale. Both clay and silt
content showed significant positive correlations with soil moisture, and sand content
showed negative correlations with deep SMC for most vegetation types. However,
soil bulk density and porosity only showed significant correlations with deep SMC
in farmland (0-220 cm) and apple orchard (60-400 cm). This result reflects that
human agricultural management measures, or other factors that result in lower soil
buck density and higher porosity conditions, can significantly improve infiltration
capacity, thus increasing deep soil moisture content.
**4.3 Implications for land use management and vegetation recovery.**
A balance between soil water availability and water utilization by plants is key
to maintaining ecosystem health, particularly in the arid and semi-arid Loess Plateau.
The implementation of the "Grain to Green Program" has effectively controlled soil
erosion (Chen et al., 2010; Wang et al., 2015a). However, according to this study,
soil desiccation occurred in almost all introduced vegetation, while higher soil
moisture content was found in native grassland and farmland (Fig. 8). These
phenomena indicate that improper selection of vegetation type is a dominant reason
for soil desiccation in this area. Thus, more attention should be paid to the selection
of vegetation types based on the interactions between soil moisture and vegetation.
Among these selected vegetation types, CK and BL caused the most serious soil
desiccation (Fig. 4 and Fig. 5); thus, these two types are especially unsuitable for
large scale plants in the study area, while SB, PG, and DP can be properly planted in





good soil moisture conditions with suitable planting density and human management
measures.
Furthermore, proper planting location should also be considered based on deep
SMC conditions. Annual average rainfall spatial variations can significantly
influence deep SMC conditions (Table 4 and Table 5). Thus, annual average rainfall
is another important factor for determining planting location. In lower rainfall zones,
vegetation enclosure and natural restoration may be good choices, while in higher
rainfall zones, shrubs and forests could be rationally arranged. Even in the same
rainfall regions, deep SMC is not evenly distributed: lower altitudes (such as a gully
bottom or lower slope) usually had higher deep SMC (Table 4 and Table 5), while
the deep SMC of native grasslands at steeper slopes was higher than that at gentle
slopes (Table 4). Thus, shrubs or trees with high water consumption capacity can be
arranged at these locations. At higher altitudes or upper slopes, where deep SMC is
lower, native grass and low moisture consuming shrubs can be arranged.
The results of this study also indicate that human agricultural management
measures can effectively improve deep SMC conditions. The SMC of farmland was
highest among the selected vegetation types (Fig. 4); even though introduced
vegetation has deep root systems, no soil desiccation was found in apple orchards
(Fig. 8). Most of the farmlands we surveyed were level terraces and back-slope level
benches with cultivation practices, while apple orchards were equipped with
artificial rainwater gathering measures. All of the agricultural measures can
significantly increase rainwater infiltration, eventually resulting in higher SMC in
these vegetation zones. Moreover, in this study, forest grasses, litter biomass, and
litter max water holding showed significant correlations with SMC (Table 5 and
Table 6). Thus, increasing land surface cover (such as crop straw coverage, mix
sowing shrub and grass) can be another effective measure for improving deep soil
moisture recharge. Likewise, considering that plant density has significant negative
correlations with SMC, vegetation control (when artificial forest and shrub are
mature, the density should be reduced according to deep soil water conditions) may





be an effective measure for helping reduce soil desiccation.
**5 Conclusions**
Based on the analysis of mean, SD, and CV of deep SMC at the watershed scale,
the results indicate that the spatial variation of deep SMC varies with soil depth and
vegetation types. In the vertical direction, the higher spatial variation of soil moisture
occurred at three depth ranges: 0-20 cm, 120-140 cm, and 480-500 cm, while in the
horizontal direction, the spatial variation in native grasses was far lower than that of
farmland, apple orchard, and introduced vegetation at comparable depths. Based on
the SMC and its variation characteristics, the SMC profile of local control natural
grassland can be divided into four layers: Ⅰ. shallow rapid change layer (0-60 cm), Ⅱ.
main rainfall infiltration layer (60-220 cm), Ⅲ. transition layer (220-400 cm), and Ⅳ.
stable layer (400-500 cm), which can reflect the influencing depths of rainfall
infiltration and evapotranspiration for SMC. Soil desiccation occurred in almost all
the vegetation types; among them, CK and BL were the most serious, indicating that
they are not suitable for large scale planting in this area. Moreover, the main rainfall
infiltration layer Ⅱ had the most serious desiccation layer. The high SMC in farmland
and apple orchard indicates that human management measures can greatly improve
deep soil moisture, even for deep-rooted apple orchards, in which no soil desiccation
was found. Although vegetation type is a dominant factor, the spatial variation
characteristic of deep soil moisture in this area is actually the combined result of
climate, vegetation, topography, soil, and human management measures. The SMC
in native grassland, which can reflect native soil moisture conditions without human
disturbance or soil moisture overconsumption, was found to be significantly related
to topography, soil traits and annual average rainfall. For introduced vegetation,
plant growth conditions, planting density, and litter water holding traits showed
significant relations with deep SMC. In farmland and orchards, human management
measures greatly increased the influence of soil traits on deep SMC, which increased





rainfall infiltration and improved deep SMC. Based on the results of this study,
proper selection of vegetation type, proper selection of planting location, and proper
landscape management measures are suggested; considering the high SMC
consumption capacity, CK and BL are unsuitable for large scale planting in the study
area, while SB, PG, and DP can be properly planted in good soil moisture conditions
with suitable planting density and human management measures. Good soil moisture
condition areas usually include higher rainfall zones and lower altitude, while human
management measures, such as macro-terrain reconstruction, artificial rainwater
gathering, increased land surface cover and vegetation density control, are effective
methods to control soil desiccation. The results of this study are of practical
significance for vegetation restoration strategies and the sustainability of restored
ecosystems.
**Acknowledgements**
This work was supported by the National Natural Science Foundation of China
(No. 41390462) and the Program for Changjiang Scholars and Innovative Research
Team in University (No. IRT_15R06). We are also grateful to the Ansai Research
Station of Soil and Water Conservation, Chinese Academy of Sciences for their
support and contributions to this fieldwork.

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
