# Peer review of "Spatial variations of deep soil moisture and the influencing factors in the Loess Plateau, China"

_Hydrology and Earth System Sciences, 2016_

## Referee Comment (RC1) · Anonymous Referee #1 · 14 Feb 2016

The goal of the study was to characterize the soil moisture under different land use and land cover for a large watershed (Ansai, Shannxi Province) in an arid region (annual precip= 500 mm) on the Loess Plateau in northern China. Field soil samplings were made at 151 spots during a 2-month period in 2014. Soil moisture content as deep as 500 cm and associated soil, vegetation and other environmental factors were measured. The authors conducted a rather thorough analysis of on the controlling factor to soil moisture by soil layer and land use types using several standard statistical methods. The researchers conclude that natural vegetation and croplands had the highest soil moisture content while introduced vegetation types have caused soil desiccation. The authors suggest that vegetation restoration in the study watershed has resulted in concerns of soil water resources depletion and this issue can explain the low productivity in planted forests. The data are valuable and findings have important implication

in practices given the large scale ecological restoration efforts in the study region. The manuscript is well written. However, a thorough read by an English native speaker will increase the readability and presentation. There are too many grammar errors and clarifications are to be addressed. Here are a few suggestions to improve the manuscript: 1. The title is misleading. The work does not address spatial variations of SMC. No maps are presented to show the differences in space across the watershed although work does examine how slope gradient, slope positions and climate (Precip) distribution result in difference in SMC. Would a map be useful to show the dominant factor controlling the overall SMC 'spatial' distribution of SMC. The authors have identified Precip and Soil Particle size (soil texture) is the major driver. But, how different is the Preci and soil across the watershed is not clear. Also, I suggest a word of watershed should be added since the paper does not address SMC for the entire Loess Plateau! 2. Which layer is considered deep soil layer? This basic concept needs to be defined clearly. 3. The ms is overly long. I suggest the authors just present key findings that are useful for illustrate the 1) overall patterns of SMC on space by soil depth, 2)contrast SMC by landuse 3)Illustrate key factors that justify the fact that the introduced vegetation had lower SMC than native grassland and crops was due to higher biomass and evapotranspiration loss NOT by other factor such as slope, aspects, soil etc. Several figs are not essential example, Fig 2 and Fig 9. Similarly reduce the number of Tables, such as Fig 7. In Table 4, only the significant correlations are needed to be reported.
* * *

---

## Referee Comment (RC2) · Anonymous Referee #2 · 16 Feb 2016

This study examines the soil moisture in deep soil profiles (up to 5 m) in a 1000 Km2 catchment of the Loess Plateau of China. The main motivation of the study is to assess the impact of different types of land management on soil water content. Three land management types are examined: i) native grasses; ii) deep root vegetation introduced as part of a large afforestation plan known as Grain for Green project; iii) cropped lands.

The number of soil profiles explored within each type of land management ranges from 10 to 25, for a total of 151 profiles. Soil samples were collected along 5 m profiles with 20 cm increments in summer 2014. Gravimetric soil water content was assessed by oven drying. Standard statistical methods have been employed to assess to what extent the deep soil moisture profiles are influenced by land vegetation types as well as other environmental factors (soil properties, terrain attributes, climate etc.).

[Figure]

Main comments

The title of the manuscript is misleading. The paper does not explore the spatial variability of the soil moisture, it rather analyses how locally observed soil moisture values are related with both natural and human induced local factors.

The manuscript is too long, it provides several details that are not relevant for the key messages of the paper. Some data could be provided as supplementary material attached to paper.

Since the main scope of the paper is to assess the effect of the vegetation on soil moisture profile, a description of the root architecture of the different vegetation species in the examined sites would facilitate the analysis of the results. At lines 15-17, page 19, the authors state that "despite the deep root system of the apple orchard…the soil moisture in the apple orchard was higher than in native grasses". But how deep is the "effective" rooting system of the apple trees? Is it really deeper than native grasses? And what about the other species? It is well known that the soil moisture profile in the inter-storm periods is influenced by the vertical distribution of the active roots. Previous studies (e.g. Laio et al., Geophysical Research Letters, 2006) showed that the vertical root distribution in water controlled ecosystems is the result on an equilibrium condition affected by the local climate and soil properties. The data provided in the paper do not prove an unbalance "between soil availability and water utilization by plants". The observed soil moisture profiles could be representative of a stationary equilibrium condition.

Other comments

Line 5-7 page 2 and Figure 2: it is not clear if the meteorological data collected during the sampling period have been exploited for the soil moisture data analyses. Apparently not. Therefore the sentence (lines 5-7) and Figure 2 can be removed. The authors should clarify to what extent the soil moisture observed in top layers could have been influenced by the rainfall events during the same sampling period.

[Figure]

Equations 1 and 2 can be removed. They describe simple metrics (depth-average soil moisture values) but are quite confusing . The same symbol SMC is used with different subscripts to describe different metrics in a way that does not appear to be consistent. From Equation 2 and the corresponding description, is not clear that SMCs represents the average soil moisture within the same type of land management at a given layer depth.

Table 2 provides details (such as Kurtosis, Skweness, K-S normality test) that are not commented in the manuscript.

Lines 10-22, page 15. The classification of the different layers is rather subjective and not supported by experimental evidences. The first layer should be influenced by both evaporation and transpiration. Not clear while the second layer is a "rainfall infiltration layer": transpiration could be significant in this layer in case of deep rooted vegetation.

Section 3.3 could be removed. It does not add information relevant for the main outcomes of the paper.

Line 15-18, page 20. It is not clear how the correlation of the soil moisture with the average annual rainfall has been computed. No data about rainfall height at the different sampling sites have been provided. The result is rather surprising. Since surface soil moisture is highly variable in time, due to evapotranspiration and rainfall events, what is the motivation of this "significant correlation"? Despite what is stated in the manuscript, Table 4 does not highlight the correlation value as "significant" (I do not see it in bold or underlined).

From pages 9-10, it seems that soil properties (particle size distribution, bulk density, porosity) have been measured only from soils cores collected from the surface. Are these properties expected to be uniform along the soil profile? Soil moisture values are significantly influenced by soil texture and organic carbon content. Do the correlations presented in Tables 4-6 refer to surface soil properties?

---

## Referee Comment (RC3) · Anonymous Referee #3 · 20 Feb 2016

The Fang et al. paper on the spatial variation of deep soil moisture in the Loess Plateau is in general well-written and it presented a very comprehensive dataset that was rarely available anywhere else. I only have a few minor comments.

1. The authors should what they meant by "deep soil moisture" early in the introduction. How deep they investigated, and the temporal and spatial scale of their experiment.

2. There are many very long paragraphs, please break them into two or more short sections.

3. I was wondering how the soil moisture was measured, did they dig a 5-m hole for each profile, or use any technology that is able to reach up to 5 m without digging a hole? If they indeed dug hole for each site, how they did it? The paper did not make it clear in all these details. It would be very impressive to dig 151, 5-m soil profile holes for any study.

4.Please check out some of my comments in the paper.

Please also note the supplement to this comment:
http://www.hydrol-earth-syst-sci-discuss.net/hess-2016-22/hess-2016-22-RC3-supplement.pdf

———————————————————

[Figure]

**Supplement:**

[revised manuscript text omitted]

---

## Referee Comment (RC4) · Anonymous Referee #4 · 21 Feb 2016

The authors of the manuscript studied the influence of soil, topography and land management on soil moisture variability to a depth of 5 m within a 1300 km2 watershed in the Loess Plateau, China. They employed the results of the analysis on parameters influencing soil moisture to evaluate controlling mechanism of soil moisture and to give some recommendation for land use management. Undisturbed soil samples were collected at 151 sites during two months (July and August 2014) from surface to a depth of 5 m at intervals of 20 cm. Gravimetric soil moisture was assessed by drying soil samples. The authors made standard statistical analysis to evaluate the influence of fifteen parameters on soil moisture content. The title is not representative of the results reported in the manuscript. The authors didn't show the spatial variation of soil moisture. The title should be more tailored on "influencing factors" rather than "spatial variation". The manuscript is too long with several repetition and some confusing

sentences. Equation 1 and 2 are not necessary. Citation should be always necessary. The need of some citation is not clear to me (i.e. at line 21 page 11). Do the authors say that tests on the distribution of data were performed by Shi et al. (2014)? In this case the authors should clearly state the origin of statistical results in table 2. Otherwise I think the citation to Shi et al. (2014) should be removed, because the need of normally distributed data to perform statistical analysis such as ANOVA was already known before Shi et al. (2014). The authors state that data were normally distributed, and then they should probably explain why they choose a non-parametric correlation test (Spearman). The authors collected soil sample during summer 2014 (two months), but they say: "Most rain occurs in the form of thunderstorms during the summer months from July to September." (lines 20-21 page 6). How they took into account the effects of rainfall and actual evapotranspiration on soil moisture dataset? The duration of the sampling campaign is a key point. In the case the measurement campaign of a single soil moisture profile at each of the 151 sites took two months, the study is questionable, because the author considered fifteen parameters without taking into account the effects of water added from thunderstorms or removed by actual evapotranspiration. The authors should clarify this point. According to data presented in table 1 the density of the solid phase of the soil varies from 2.37 to 2.47 Mg m-3. How the authors measured this parameter? Why the authors decided to employ a variable density of the solid phase? A constant solid phase density would establish a linear relation between porosity and soil bulk density. How many soil samples for measurement of particle size distribution were collected at each site? Were the soil samples collected on soil surface or along the soil profile? In some cases the authors drawn conclusions from results of statistical analysis, but in the discussion they didn't give any explanation on the hydrological processes that could have led to such results. Since any influence was observed in the upper layers, why soil moisture between 4 and 5 m depth below David peach should be influenced by grass biomass? Same question should be answered for the influence of litter biomass below apple orchard. Finally, the authors should change "buck density" to "bulk density" and "organic" to "organic matter". Pay attention to the

use of "infiltration", sometimes was used instead of "storage".

---

## Author Comment (AC1) · 31 Mar 2016

We thank reviewer for the detailed comments. We have gone through all the comments and will amend the original manuscript base on the suggestions and comments. In the following pages we provide brief answers to the reviews comments and we will make corresponding changes when we receive the editor decision.

Reviewer: The researchers conclude that natural vegetation and croplands had the highest soil moisture content while introduced vegetation types have caused soil desiccation. The authors suggest that vegetation restoration in the study watershed has resulted in concerns of soil water resources depletion and this issue can explain the low productivity in planted forests. The data are valuable and findings have important implication in practices given the large-scale ecological restoration efforts in the study

region.

Authors: Discussion of the deep soil moisture condition under different vegetation types and its control mechanism at watershed scale is indeed a valuable and challenging task. Thank you very much for your encouragement. We will carefully amend the manuscript based on the comments that you provided.

Reviewer: The manuscript is well written. However, a thorough read by an English native speaker will increase the readability and presentation. There are too many grammar errors and clarifications are to be addressed.

Authors: In order to increase the readability of the manuscript, we will invite a native English speaker to revise the language of our manuscript.

Reviewer: The title is misleading. The work does not address spatial variations of SMC. No maps are presented to show the differences in space across the watershed although work does examine how slope gradient, slope positions and climate (Precip) distribution result in difference in SMC.

Authors: Based on the collective suggestions from all the reviewers, we will adjust the title in the revised manuscript, and remove the "spatial variations" from the title.

Reviewer: The authors have identified Precipitation and Soil Particle size (soil texture) is the major driver. But, how different is the Precipitation and soil across the watershed is not clear. Also, I suggest a word of watershed should be added since the paper does not address SMC for the entire Loess Plateau!

Authors: We agree it is necessary to clarity the differences of precipitation and soil particle size across the watershed. Thus, we will add relevant contents in the revised manuscript. Besides, following the reviewer's suggestion, we will add a word of "watershed" in the title as well.

Reviewer: Which layer is considered deep soil layer? This basic concept needs to be defined clearly.

[Figure]

Authors: In this study, deep soil layer refers to the layer whose soil moisture can seldom be influenced by rainfall infiltration and evaporation during a certain time period. We will define "deep soil layer" clearly in the Introduction section of revised manuscript.

Reviewer: The manuscript is overly long. I suggest the authors just present key findings that are useful for illustrate the 1) overall patterns of SMC on space by soil depth, 2) contrast SMC by land use 3) Illustrate key factors that justify the fact that the introduced vegetation had lower SMC than native grassland and crops was due to higher biomass and evapotranspiration loss NOT by other factor such as slope, aspects, soil etc. Several figs are not essential example, Fig 2 and Fig 9. Similarly reduce the number of Tables, such as Fig 7. In Table 4, only the significant correlations are needed to be reported.

Authors: Following the reviewer's suggestion, we will exam the manuscript carefully and remove/condense some sections, especially for figures, and tables that are of less relevance with key findings.

---

## Author Comment (AC2) · 31 Mar 2016

We thank reviewer for the detailed comments. We have gone through all the comments and will amend the original manuscript base on the suggestions and comments. In the following pages we provide brief answers to the reviews comments and we will make corresponding changes when we receive the editor decision.

Reviewer: The title of the manuscript is misleading. The paper does not explore the spatial variability of the soil moisture, it rather analyses how locally observed soil moisture values are related with both natural and human induced local factors.

Authors: Following the reviewer's suggestion, we will revise the title in the revised manuscript, and the "Spatial variations" will be removed.

[Figure]

Reviewer: The manuscript is too long, it provides several details that are not relevant for the key messages of the paper. Some data could be provided as supplementary material attached to paper.

Authors: Following the reviewer's suggestion, we will exam the manuscript carefully and remove/condense some sections, especially for figures, and tables that are of less relevance with key findings. We will move these figures and tables as supplementary materials.

Reviewer: Since the main scope of the paper is to assess the effect of the vegetation on soil moisture profile, a description of the root architecture of the different vegetation species in the examined sites would facilitate the analysis of the results. At lines 15-17, page 19, the authors state that "despite the deep root system of the apple orchard . . .the soil moisture in the apple orchard was higher than in native grasses". But how deep is the "effective" rooting system of the apple trees? Is it really deeper than native grasses? And what about the other species?

Authors: We agree a description of the root architecture of the different vegetation species can really help facilitate the analysis of the results. However, it is nearly logistically impossible for us to dig out the whole rooting system of all the plants in the 151 sampling sites. Thus, root architectures of the eight vegetation species in the study area will be obtained through other publications. In the revised manuscript, we will display the root architecture of different vegetation types.

Reviewer: It is well known that the soil moisture profile in the inter-storm periods is influenced by the vertical distribution of the active roots. Previous studies (e.g. Laio et al., Geophysical Research Letters, 2006) showed that the vertical root distribution in water controlled ecosystems is the result on an equilibrium condition affected by the local climate and soil properties. The data provided in the paper do not prove an unbalance "between soil availability and water utilization by plants". The observed soil moisture profiles could be representative of a stationary equilibrium condition.

Authors: We agree the vertical distribution of the active roots can influence soil moisture profile in the inter-storm periods. However, based on the EM50 dynamic monitoring data (shallow-root system native grasses and deep-root system Caragana korshinskii), we can conclude during the sampling period (July 10 - August 6), root distribution mainly influenced the soil moisture no deeper than 80cm, and deeper soil moisture (80-500cm) was not influenced. Thus, in the revised manuscript we will add the EM50 dynamic monitoring data, and clarify the roots influences on soil moisture.

Reviewer: Line 5-7 page 2 and Figure 2: it is not clear if the meteorological data collected during the sampling period have been exploited for the soil moisture data analyses. Apparently not. Therefore the sentence (lines 5-7) and Figure 2 can be removed. The authors should clarify to what extent the soil moisture observed in top layers could have been influenced by the rainfall events during the same sampling period.

Authors: The reviewer is correct, we didn't analyze the meteorological data collected during the sampling period in terms of soil moisture data analyeses, it was used to illustrate the climate condition of sampling period (July 10 - August 6). Thus, we will remove the sentence (lines 5-7) and Figure 2 in the revised manuscript. Besides, based on the observations, the rainfall events during the sampling period influenced soil moisture no deeper than 80cm. This can be verified by our EM50 dynamic monitoring data which will be added in the revised manuscript.

Reviewer: Equations 1 and 2 can be removed. They describe simple metrics (depth-average soil moisture values) but are quite confusing. The same symbol SMC is used with different subscripts to describe different metrics in a way that does not appear to be consistent. From Equation 2 and the corresponding description, is not clear that SMCs represents the average soil moisture within the same type of land management at a given layer depth.

Authors: Following the reviewer's suggestion, we will remove Equations 1 and 2 in the

revised manuscript.

Reviewer: Table 2 provides details (such as Kurtosis, Skweness, K-S normality test) that are not commented in the manuscript.

Authors: Following the reviewer's suggestion, the detailed description of Table 2 (including Kurtosis, Skweness, K-S normality test) will be added in the revised manuscript.

Reviewer: Lines 10-22, page 15. The classification of the different layers is rather subjective and not supported by experimental evidences. The first layer should be influenced by both evaporation and transpiration. Not clear while the second layer is a "rainfall infiltration layer": transpiration could be significant in this layer in case of deep-rooted vegetation.

Authors: We agree the classification of the different soil layers by only considering soil moisture variation in native grassland is subjective. In the revised manuscript, we will further improve the classification method based on experimental evidence and take transpiration of deep-rooted vegetation into consideration.

Reviewer: Section 3.3 could be removed. It does not add information relevant for the main outcomes of the paper.

Authors: As suggested, we will remove the section 3.3 due to its loose relationships with the main points of the manuscript.

Reviewer: Line 15-18, page 20. It is not clear how the correlation of the soil moisture with the average annual rainfall has been computed. No data about rainfall height at the different sampling sites have been provided. The result is rather surprising. Since surface soil moisture is highly variable in time, due to evapotranspiration and rainfall events, what is the motivation of this "significant correlation"? Despite what is stated in the manuscript, Table 4 does not highlight the correlation value as "significant" (I do not see it in bold or underlined).

Authors: The average annual rainfall (2006-2013) was provided by 29 rain gauges in or around the Ansai watershed, and the Ordinary Kriging method was performed by ArcGIS10.0 to obtain the average annual rainfall at each sampling site. In order to further illustrate this result, we will add a distribution map of rainfall in the supplement documents. Besides, we further checked Table 4, and found that all the bold style in this table is missing. There must be something wrong when we uploaded the manuscript. We apologize for this oversight and we will correct this table in the revised manuscript.

Reviewer: From pages 9-10, it seems that soil properties (particle size distribution, bulk density, porosity) have been measured only from soils cores collected from the surface. Are these properties expected to be uniform along the soil profile? Soil moisture values are significantly influenced by soil texture and organic carbon content. Do the correlations presented in Tables 4-6 refer to surface soil properties?

Authors: Yes, all the soil properties have been measured only from soils cores collected from the surface (0-20cm). In loess Plateau, loess soil thickness in this area ranges from 30-80 m, and groundwater below this depth can merely influence deep soil moisture that are available for plants growth. Thus, the deep soil moisture in this region is mainly determined by land surface rainfall infiltration and evapotranspiration. As we know, surface soil properties (such as particle size distribution, bulk density, porosity, and organic carbon content) are usually more important in influencing surface rainfall infiltration and evaporation than deep soil properties, thus in this study we mainly analyzed surface soil properties influence on deep soil moisture. In the revised manuscript we will further explain related reasons.

---

## Author Comment (AC3) · 31 Mar 2016

We thank reviewer for the constructive comments. We have gone through all the comments and will amend the original manuscript base on the suggestions and comments. In the following pages we provide brief answers to the reviews comments and we will make corresponding changes when we receive the editor decision.

Reviewer: The Fang et al. paper on the spatial variation of deep soil moisture in the Loess Plateau is in general well-written and it presented a very comprehensive dataset that was rarely available anywhere else.

Authors: Discussion of the deep soil moisture condition under different vegetation types and its control mechanism at watershed scale is indeed a valuable and challenging task. Thank you very much for your encouragement. We will carefully amend the

manuscript based on the comments that the reviewers provided.

Reviewer: The authors should clarify what they meant by "deep soil moisture" early in the introduction. How deep they investigated, and the temporal and spatial scale of their experiment.

Authors: As suggested by the reviewer, we will define "deep soil layer" clearly in the introduction section of revised manuscript, and describe the investigation depth, temporal and spatial scale of our experiment in detail.

Reviewer: There are many very long paragraphs, please break them into two or more short sections. (page4,paragraph 2)

Authors: As suggested by the reviewer, we will break down the long paragraphs into short sections.

Reviewer: I was wondering how the soil moisture was measured, did they dig a 5-m hole for each profile, or use any technology that is able to reach up to 5 m without digging a hole? If they indeed dug hole for each site, how they did it? The paper did not make it clear in all these details. It would be very impressive to dig 151, 5-m soil profile holes for any study.

Authors: Actually, the soil samples in depth of 0–5m were taken by a soil drill (5 cm in diameter) with 20-cm increment. It is indeed a challenging task in logistics and therefore the data are quite valuable. We will clearly describe the sampling details in the revised manuscript.

Reviewer: In page 3, line 7-9 only include a few representative references, this is too many.

Authors: As suggested by the reviewer, we will further exam the references, and only retain a few representative ones.

Reviewer: In page4, line22. Should you simply use deep soil moisture (DSM) only?

[Figure]

This is the term you used in your title.

Authors: Yes, "deep soil moisture (DSM)" is more accurate than "deep soil moisture content (SMC)", thus we will replace it in the revised manuscript.

Reviewer: "whole" in page 5, line 4, "According to previous studies, factors that control deep SMC variations are different under three land management types: native vegetation with a shallow root system, introduced vegetation with a deep root system, and vegetation with agricultural management measures (Jia et al., 2013; Jia and Shao, 2014; Yang et al., 2012b; Yang et al., 2014a)." in page 5, line27-29, and "The Ansai watershed is located on a warm forest steppe" in page 6, line 23 should be deleted.

Authors: Following the reviewer's suggestion, we will delete these less relevant sentences in the revised manuscript.

Reviewer: In page 7,line 4.You should include the boundary of Shanxi province since you mentioned it in your description.

Authors: Following the reviewer's suggestion, we will add the boundary of Shanxi province in Figure 1 in the revised manuscript.

Reviewer: In page 13, line 1. You don't need to include the legends in the graphs since there is only one category.

Authors: As suggested by the reviewer, we will delete the unnecessary legend in the graphs in the revised manuscript.

Reviewer: In page 24, line 8. I don't think you need to count to two digits, simply 80%, 68% etc...

Authors: As suggested by the reviewer, we will only reserve integer digits in the revised manuscript.

---

## Author Comment (AC4) · 31 Mar 2016

We thank reviewer for the detailed comments. We have gone through all the comments and will amend the original manuscript base on the suggestions and comments. In the following pages we provide brief answers to the reviews comments and we will make corresponding changes when we receive the editor decision.

Reviewer: The title is not representative of the results reported in the manuscript. The authors didn't show the spatial variation of soil moisture. The title should be more tailored on "influencing factors" rather than "spatial variation".

Authors: Considering the collective suggestions from all four reviewers, we will revise the title in the revised manuscript, and make the title more tailored on "influencing factors".

Reviewer: The manuscript is too long with several repetition and some confusing sentences.

Authors: Following the reviewer's suggestion, we will exam the manuscript carefully and remove some sections that are less relevant with our key findings. Besides, we will invite a native English speaker to revise the language of our manuscript to increase the readability of the manuscript.

Reviewer: Equation 1 and 2 are not necessary.

Authors: Following the reviewer's suggestion, we will delete equation 1 and 2.

Reviewer: Citation should be always necessary. The need of some citation is not clear to me (i.e. at line 21 page 11). Do the authors say that tests on the distribution of data were performed by Shi et al. (2014)? In this case the authors should clearly state the origin of statistical results in table 2. Other- wise I think the citation to Shi et al. (2014) should be removed, because the need of normally distributed data to perform statistical analysis such as ANOVA was already known before Shi et al. (2014).

Authors: We agree some citations in this manuscript may be inaccurate; we will carefully exam all the citations in the manuscript to ensure their accuracy, and remove the unnecessary ones.

Reviewer: The authors state that data were normally distributed, and then they should probably explain why they choose a non-parametric correlation test (Spearman).

Authors: The data were normally distributed; however, significant correlations exist in SMC at different soil depth ranges (Figure 7). So, we chose a non-parametric correlation test (Spearman) in this manuscript. In the revised manuscript, we will further clarify this point.

Reviewer: The authors collected soil sample during summer 2014 (two months), but they say: "Most rain occurs in the form of thunderstorms during the summer months from July to September." (lines 20-21 page 6). How they took into account the effects

of rainfall and actual evapotranspiration on soil moisture dataset? The duration of the sampling campaign is a key point. In the case the measurement campaign of a single soil moisture profile at each of the 151 sites took two months, the study is questionable, because the author considered fifteen parameters without taking into account the effects of water added from thunderstorms or removed by actual evapotranspiration. The authors should clarify this point.

Authors: Actually, the exact sampling period is from July 10 to August 6. Based on field observation and EM50 dynamic monitoring data, the rainfall events and evapotranspiration influenced soil moisture no deeper than 80cm, thus we consider the deep soil moisture (80-500cm) is seldom influenced by rainfall events and evapotranspiration during the sampling period. Besides, the main objective of this manuscript is to exam the variation of deep soil moisture and its influencing factors, the surface layer soil moisture variation is only considered as a comparison. We will further clarify this point in the revised manuscript, and EM50 dynamic monitoring data will be added to verify this point.

Reviewer: According to data presented in Table 1 the density of the solid phase of the soil varies from 2.37 to 2.47 Mg m-3. How the authors measured this parameter? Why the authors decided to employ a variable density of the solid phase? A constant solid phase density would establish a linear relation between porosity and soil bulk density.

Authors: Actually, the data "porosity" presented in Table 1 is "capillary porosity" not "soil total porosity". Capillary porosity was calculated by solid phase density and bulk density. To calculate this parameter, undistorted soil cores were collected in metal cylinders (diameter 5 cm, length 5 cm) at each sampling site, and then capillary porosity were measured by "cylinder soak method". In the revised manuscript, we will change "porosity" to "capillary porosity" in order to eliminate confusion.

Reviewer: In some cases the authors drawn conclusions from results of statistical analysis, but in the discussion they didn't give any explanation on the hydrological

processes that could have led to such results. Since any influence was observed in the upper layers, why soil moisture between 4 and 5 m depth below David peach should be influenced by grass biomass? Same question should be answered for the influence of litter biomass below apple orchard.

Authors: As suggested by the reviewer, we will exam results of statistical analysis and provide explanations on the hydrological processes that could have led to such results in the revised manuscript.

Reviewer: the authors should change "buck density" to "bulk density" and "organic" to "organic matter". Pay attention to the use of "infiltration", sometimes was used instead of "storage".

Authors: Following the reviewer's suggestion, we will exam the entire manuscript and change all the "buck density" to "bulk density" and "organic" to "organic matter". We will also carefully check the use of "infiltration".

————————————————————

---

## Author Response (AR1)

**Responses to the Reviewers:**

We thank the editor and reviewers very much for the time they spent evaluating our manuscript and providing constructive comments. Their detailed comments inspired us to improve the quality of this manuscript. We have carefully amended our original manuscript based on the suggestions and comments. Our detailed responses are provided below.

**Responses to Reviewer #1:**

*Reviewer: The researchers conclude that natural vegetation and croplands had the highest soil moisture content while introduced vegetation types have caused soil desiccation. The authors suggest that vegetation restoration in the study watershed has resulted in concerns of soil water resources depletion and this issue can explain the low productivity in planted forests. The data are valuable and findings have important implication in practices given the large-scale ecological restoration efforts in the study region.*

**Authors:** Investigating the deep soil moisture (DSM) dynamics under different vegetation types and its control mechanism at the watershed scale is indeed a valuable and challenging task. Thank you very much for your encouragement. We have carefully amended the manuscript based on the comments that you provided.

*Reviewer: The manuscript is well written. However, a thorough read by an English native speaker will increase the readability and presentation. There are too many grammar errors and clarifications are to be addressed.*

**Authors:** We have invited a native English speaker to revise the language of our manuscript to increase its readability.

*Reviewer: The title is misleading. The work does not address spatial variations of SMC. No maps are presented to show the differences in space across the watershed although work does examine how slope gradient, slope positions and climate (Precip) distribution result in difference in SMC.*

**Authors:** We have adjusted the title in the revised manuscript based on the collective suggestions from all the reviewers (please refer to the title section on page 1).

*Reviewer: The authors have identified Precipitation and Soil Particle size (soil texture)*

*is the major driver. But, how different is the Precipitation and soil across the watershed is not clear. Also, I suggest a word of watershed should be added since the paper does not address SMC for the entire Loess Plateau!*

**Authors:** We agree that clarifying the differences in precipitation and soil particle size across the watershed is necessary. Thus, we have added relevant content to the revised manuscript and supplemental materials (please refer to lines 2-4 on page 7, lines 7-9 on page 7, and Figures S1 and S2 in the supplemental material). Following the reviewer's suggestion, we have added the word "watershed" in the title (please refer to the title section on page 1).

*Reviewer: Which layer is considered deep soil layer? This basic concept needs to be defined clearly.*

**Authors:** In this study, we define a deep soil layer as a layer where the soil moisture is not sensitive to daily evapotranspiration and regular rainfall events. We have defined "deep soil layer" clearly in the Introduction section of the revised manuscript (please refer to lines 15-26 on page 3 and lines 1-2 on page 4).

*Reviewer: The manuscript is overly long. I suggest the authors just present key findings that are useful for illustrate the 1) overall patterns of SMC on space by soil depth, 2) contrast SMC by land use 3) Illustrate key factors that justify the fact that the introduced vegetation had lower SMC than native grassland and crops was due to higher biomass and evapotranspiration loss NOT by other factor such as slope, aspects, soil etc. Several figs are not essential example, Fig 2 and Fig 9. Similarly reduce the number of Tables, such as Fig 7. In Table 4, only the significant correlations are needed to be reported.*

**Authors:** Following the reviewer's suggestion, we have checked the manuscript carefully and removed/condensed some content that was less relevant to the key findings, such as Figure 2 (lines 1-2 on page 12), Figure 6 (lines 14-21 on page 22 and lines 1-11 on page 23), Section 3.3 (lines 12-26 on page 23 and lines 1-2 on page 24), and Figure 9 (lines 1-10 on page 35).

**Response to reviewer #2:**

*Reviewer: The title of the manuscript is misleading. The paper does not explore the spatial variability of the soil moisture, it rather analyses how locally observed soil moisture values are related with both natural and human induced local factors.*

**Authors:** We have adjusted the title of the revised manuscript based on the collective suggestions from all the reviewers (please refer to the title section on page 1).

**Reviewer:** *The manuscript is too long, it provides several details that are not relevant for the key messages of the paper. Some data could be provided as supplementary material attached to paper.*

**Authors:** Following the reviewer's suggestion, we have checked through the manuscript carefully and removed/condensed some content that was less relevant to the key findings, such as Figure 2 (lines 1-2 on page 12), Figure 6 (lines 14-21 on page 22 and lines 1-11 on page 23), Section 3.3 (lines 12-26 on page 23 and lines 1-2 on page 24), and Figure 9 (lines 1-10 on page 35). Additionally, we provided the annual average rainfall distribution data, soil particle composition distribution data, and meteorological data as supplementary materials, which are attached to this manuscript (please refer to Section S1, S2, and S3 in the supplemental materials).

**Reviewer:** *Since the main scope of the paper is to assess the effect of the vegetation on soil moisture profile, a description of the root architecture of the different vegetation species in the examined sites would facilitate the analysis of the results. At lines 15-17, page 19, the authors state that "despite the deep root system of the apple orchard . . .the soil moisture in the apple orchard was higher than in native grasses". But how deep is the "effective" rooting system of the apple trees? Is it really deeper than native grasses? And what about the other species?*

**Authors:** We agree that a description of the root architecture of the different vegetation species can facilitate an analysis of the results. However, digging out the entire rooting systems of all the plants in the 151 sampling sites would be nearly impossible. Thus, root architecture information of the eight vegetation species in the study area has been obtained from other publications. We added the root architecture information of different vegetation types to the revised manuscript (see Table 1 on page 9).

**Reviewer:** *It is well known that the soil moisture profile in the inter-storm periods is influenced by the vertical distribution of the active roots. Previous studies (e.g. Laio et al., Geophysical Research Letters, 2006) showed that the vertical root distribution in water controlled ecosystems is the result on an equilibrium condition affected by the local climate and soil properties. The data provided in the paper do not prove an unbalance "between soil availability and water utilization by plants". The observed soil moisture profiles could be representative of a stationary equilibrium condition.*

**Authors:** We agree that the vertical distribution of the active roots can influence the soil moisture profile during inter-storm periods. However, EM50 dynamic monitoring data (shallow-root system native grasses and deep-root system *Caragana korshinskii*) indicated that no significant changes in deeper soil moisture (80-500 cm) occurred during the sampling period (July 10 - August 6). In contrast to surface soil moisture, which is greatly influenced by vegetation roots, the DSM is relatively stable over a short period (such as a month) and is probably determined by long-term moisture replenishment and consumption. We only explored stationary DSM data in this study; however, we can still demonstrate an imbalance between soil availability and water utilization by plants by comparing DSM under different vegetation types. For example, soil desiccation in introduced vegetation supported a long-term imbalance between rainfall infiltration and water consumption by plant root systems. We have added these EM50 dynamic monitoring data and clarified the influence of the roots on soil moisture in the revised manuscript (see section 3.1 on pages 14-15).

*Reviewer: Line 5-7 page 2 and Figure 2: it is not clear if the meteorological data collected during the sampling period have been exploited for the soil moisture data analyses. Apparently not. Therefore the sentence (lines 5-7) and Figure 2 can be removed. The authors should clarify to what extent the soil moisture observed in top layers could have been influenced by the rainfall events during the same sampling period.*

**Authors:** No, we did not analyze the meteorological data that were collected during the sampling period in terms of soil moisture data analyses; these data were used to illustrate the climate conditions of the sampling period (July 10 - August 6). In the revised manuscript, we have further exploited the meteorological data and moved Figure 2 and relevant content to the supplemental materials (see lines 16-18 on page 11, lines 1-2 on page 12, and section S3 in the supplemental materials). According to our EM50 dynamic monitoring data, the rainfall events during the sampling period influenced soil moisture no deeper than 80 cm; thus, we mainly analyzed the soil moisture at depths of 80-500 cm in the revised manuscript (see section 3.1 on pages 14-15).

*Reviewer: Equations 1 and 2 can be removed. They describe simple metrics (depth-average soil moisture values) but are quite confusing. The same symbol SMC is used with different subscripts to describe different metrics in a way that does not appear to be consistent. From Equation 2 and the corresponding description, is not clear that SMCs represents the average soil moisture within the same type of land management at a given layer depth.*

**Authors:** Following the reviewer's suggestion, we have removed Equations 1 and 2 from the revised manuscript (see lines 6-13 on page 23).

*Reviewer: Table 2 provides details (such as Kurtosis, Skweness, K-S normality test) that are not commented in the manuscript.*

**Authors:** Following the reviewer's suggestion, a detailed description of Table 2

(including the kurtosis, skewness, and K-S normality tests) has been added to the revised manuscript (see lines 16-20 on page 15).

*Reviewer: Lines 10-22, page 15. The classification of the different layers is rather subjective and not supported by experimental evidences. The first layer should be influenced by both evaporation and transpiration. Not clear while the second layer is a "rainfall infiltration layer": transpiration could be significant in this layer in case of deep-rooted vegetation.*

**Authors:** We agree that the classification of the different soil layers by only considering soil moisture variations in native grassland is subjective. In the revised manuscript, we have further adjusted the classification based on EM50 dynamic monitoring data and removed the 0-80-cm soil moisture layer because this layer is not relevant in terms of deep soil moisture. Additionally, we have included the transpiration of deep-rooted vegetation (see lines 6-22 on page 19 and lines 1-20 on page 20).

*Reviewer: Section 3.3 could be removed. It does not add information relevant for the main outcomes of the paper.*

**Authors:** As suggested, we have removed section 3.3 because it is not closely related to the main points of the manuscript (see lines 12-26 on page 23 and lines 1-2 on page 24).

*Reviewer: Line 15-18, page 20. It is not clear how the correlation of the soil moisture with the average annual rainfall has been computed. No data about rainfall height at the different sampling sites have been provided. The result is rather surprising. Since surface soil moisture is highly variable in time, due to evapotranspiration and rainfall events, what is the motivation of this "significant correlation"? Despite what is stated in the manuscript, Table 4 does not highlight the correlation value as "significant" (I do not see it in bold or underlined).*

**Authors:** Actually, no rainfall monitoring was conducted at any of the sampling sites. The average annual rainfall (2006-2013) was provided by 29 rain gauges in or around the Ansai watershed, and the Inverse Distance Weighted (IDW) interpolation method was performed by ArcGIS 10.0 to obtain the average annual rainfall at each sampling site (see lines 18-21 on page 11). We have added a distribution map of the average annual rainfall in the supplemental materials to further illustrate this approach (see Figure S1 in the supplemental materials). Additionally, we checked Table 4 and found that the bold style in this table was missing. We apologize for this oversight, and we have corrected this table in the revised manuscript (see Table 5 on page 28).

***Reviewer:*** *From pages 9-10, it seems that soil properties (particle size distribution, bulk density, porosity) have been measured only from soils cores collected from the surface. Are these properties expected to be uniform along the soil profile? Soil moisture values are significantly influenced by soil texture and organic carbon content. Do the correlations presented in Tables 4-6 refer to surface soil properties?*

**Authors:** Yes, all the soil properties were measured only from soil cores that were collected from the surface (0-20 cm). The loess soil thickness in the Loess Plateau ranges from 30 to 80 m, and groundwater below this depth cannot influence deep soil moisture that is available for plant growth. Thus, the deep soil moisture in this region is mainly determined by land surface rainfall infiltration and evapotranspiration. Although soil properties may be different along the soil profile, surface soil properties (such as the particle size distribution, bulk density, porosity, and organic carbon content) usually have a greater influence on surface rainfall infiltration and evaporation compared to deep soil properties, and measuring the soil properties (especially bulk density) from the 0-500-cm layer at 151 sampling spots is nearly impossible. Thus, we mainly focused on analyzing the surface soil properties' influence on deep soil moisture. We have further explained the reason for this approach in the revised manuscript (see lines 10-13 on page 40).

**Response to reviewer #3:**

***Reviewer:*** *The Fang et al. paper on the spatial variation of deep soil moisture in the Loess Plateau is in general well-written and it presented a very comprehensive dataset that was rarely available anywhere else.*

**Authors:** Thank you very much for your encouragement. Measuring deep soil moisture at depths of 0-500 cm is a very challenging but valuable task. We have carefully amended the manuscript based on your comments.

***Reviewer:*** *The authors should clarify what they meant by "deep soil moisture" early in the introduction. How deep they investigated, and the temporal and spatial scale of their experiment.*

**Authors:** As suggested by the reviewer, we have defined "deep soil layer" clearly in the introduction section of the revised manuscript (see lines 15-26 on page 3 and lines 1-2 on page 4) and described the investigation depth and temporal and spatial scales of our experiment in detail (lines 5-9 on page 11).

***Reviewer:*** *There are many very long paragraphs, please break them into two or more*

*short sections. (page4, paragraph 2)*

**Authors:** As suggested by the reviewer, we have broken down long paragraphs into short sections (see lines 23-24 on page 5 and lines 2-3 on page 40).

*Reviewer: I was wondering how the soil moisture was measured, did they dig a 5-m hole for each profile, or use any technology that is able to reach up to 5 m without digging a hole? If they indeed dug hole for each site, how they did it? The paper did not make it clear in all these details. It would be very impressive to dig 151, 5-m soil profile holes for any study.*

**Authors:** Actually**,** the soil samples at depths of 0–500 cm were collected by a soil drill (5 cm in diameter) with 20-cm increments (see the following figure).

[Figure]

Collecting these data is indeed a challenging logistical task, so these data are quite valuable. We have clearly described the sampling details in the revised manuscript (see lines 5-9 on page 11).

*Reviewer: In page 3, line 7-9 only include a few representative references, this is too many.*

**Authors:** As suggested by the reviewer, we have further checked the references and only retained a few representative citations in the revised manuscript (see lines 12-15 on page 3).

*Reviewer: In page4, line22. Should you simply use deep soil moisture (DSM) only? This is the term you used in your title.*

**Authors:** Yes, "deep soil moisture (DSM)" is more accurate than "deep soil moisture content (SMC)"; thus, we have replaced this term throughout the revised manuscript.

*Reviewer: "whole" in page 5, line 4, "According to previous studies, factors that control deep SMC variations are different under three land management types: native vegetation with a shallow root system, introduced vegetation with a deep root system, and vegetation with agricultural management measures (Jia et al., 2013; Jia and Shao, 2014; Yang et al., 2012b; Yang et al., 2014a)." in page 5, line27-29, and "The Ansai watershed is located on a warm forest steppe" in page 6, line 23 should be deleted.*

**Authors:** Following the reviewer's suggestion, we have deleted these less relevant sentences in the revised manuscript (see line 20 on page 5, lines 15-19 on page 6, and lines 20-21 on page 7).

*Reviewer: In page 7, line 4. You should include the boundary of Shanxi province since you mentioned it in your description.*

**Authors:** Following the reviewer's suggestion, we have added the boundary of Shanxi province in Figure 1 in the revised manuscript (see Figure 1 on page 8).

*Reviewer: In page 13, line 1. You don't need to include the legends in the graphs since there is only one category.*

**Authors:** As suggested by the reviewer, we have deleted the unnecessary legend in the graphs in the revised manuscript (see lines 1-3 on page 17).

*Reviewer: In page 24, line 8. I don't think you need to count to two digits, simply 80%, 68% etc...*
**Authors:** As suggested by the reviewer, we have reported integer digits in the revised manuscript (see lines 3-6 on page 32).

**Response to reviewer #4:**

***Reviewer:*** *The title is not representative of the results reported in the manuscript. The authors didn't show the spatial variation of soil moisture. The title should be more tailored on "influencing factors" rather than "spatial variation".*

**Authors:** Considering the collective suggestions from all four reviewers, we have revised the title of the revised manuscript (see the title section on page 1).

***Reviewer:*** *The manuscript is too long with several repetition and some confusing sentences.*

**Authors:** Following the reviewer's suggestion, we have removed some content that was less relevant to the key findings, such as Figure 2 (lines 1-2 on page 12), Figure 6 (lines 14-21 on page 22 and lines 1-11 on page 23), Section 3.3 (lines 12-26 on page 23 and lines 1-2 on page 24), and Figure 9 (lines 1-10 on page 35). In addition, we have invited a native English speaker to revise the language of our manuscript to increase its readability.

***Reviewer:*** *Equation 1 and 2 are not necessary.*

**Authors:** Following the reviewer's suggestion, we have deleted Equations 1 and 2 (see lines 6-13 on page 13).

***Reviewer:*** *Citation should be always necessary. The need of some citation is not clear to me (i.e. at line 21 page 11). Do the authors say that tests on the distribution of data were performed by Shi et al. (2014)? In this case the authors should clearly state the origin of statistical results in table 2. Other- wise I think the citation to Shi et al. (2014) should be removed, because the need of normally distributed data to perform statistical analysis such as ANOVA was already known before Shi et al. (2014).*

**Authors:** We agree that some citations in this manuscript may be unnecessary; we have carefully checked all the citations in the manuscript to ensure their accuracy and removed unnecessary citations (see lines 12-15 on page 3 and line 20 on page 15).

***Reviewer:*** *The authors state that data were normally distributed, and then they should probably explain why they choose a non-parametric correlation test (Spearman).*

**Authors:** The data were normally distributed; however, significant correlations existed in the soil moisture content at different soil depth ranges (see Figure 7 on page 24). Thus, we chose a non-parametric correlation test (Spearman), which we have further clarified in the revised manuscript (see lines 5-7 on page 27).

***Reviewer:*** *The authors collected soil sample during summer 2014, but they say: "Most rain occurs in the form of thunderstorms during the summer months from July to September." (lines 20-21 page 6). How they took into account the effects of rainfall and actual evapotranspiration on soil moisture dataset? The duration of the sampling campaign is a key point. In the case the measurement campaign of a single soil moisture profile at each of the 151 sites took two months, the study is questionable, because the author considered fifteen parameters without taking into account the effects of water added from thunderstorms or removed by actual evapotranspiration. The authors should clarify this point.*

**Authors:** Actually, the duration of the sampling campaign was 28 days (from July 10 to August 6). According to field observation and EM50 dynamic monitoring data, the rainfall events and evapotranspiration influenced soil moisture no deeper than 80 cm, so we consider that deep soil moisture (80-500 cm) was seldom influenced by rainfall events and evapotranspiration during the sampling period. Additionally, the main objective of this manuscript is to examine variations in the deep soil moisture and its influencing factors, so we have removed our analysis of the 0-80-cm soil moisture data. We have further clarified this point in the revised manuscript, and EM50 dynamic monitoring data have been added to verify this point (see section 3.1 on pages 14-15).

***Reviewer:*** *According to data presented in Table 1 the density of the solid phase of the soil varies from 2.37 to 2.47 Mg m⁻³. How the authors measured this parameter? Why the authors decided to employ a variable density of the solid phase? A constant solid phase density would establish a linear relation between porosity and soil bulk density.*

**Authors:** Actually, "porosity" in Table 1 is "capillary porosity", not "soil total porosity". The capillary porosity was calculated from the solid phase density and bulk density. Undisturbed soil cores were collected in metal cylinders (diameter of 5 cm and length of 5 cm) at each sampling site, and then the capillary porosity was measured by the "cylinder soak method". We have changed "porosity" to "capillary porosity" in the revised manuscript to avoid confusion (see lines 9-12 on page 12).

***Reviewer:*** *In some cases the authors drawn conclusions from results of statistical analysis, but in the discussion they didn't give any explanation on the hydrological processes that could have led to such results. Since any influence was observed in the upper layers, why soil moisture between 4 and 5 m depth below David peach should be influenced by grass biomass? Same question should be answered for the influence of litter biomass below apple orchard.*

**Authors:** As suggested by the reviewer, we have checked the results of the statistical analysis and provided explanations for the hydrological processes that could have led to such results in the revised manuscript (see lines 26-29 on page 38). The deep soil moisture below David peach trees and apple orchards had significantly positive relationships with the upper layer grass biomass and litter biomass, probably because thick litter and forest grasses can reduce surface runoff, which may help retain more rainfall for infiltration into deep soil layers. In addition, these factors can reduce soil evaporation, which may decrease DSM consumption.

***Reviewer****: the authors should change "buck density" to "bulk density" and "organic" to "organic matter". Pay attention to the use of "infiltration", sometimes was used instead of "storage".*

**Authors:** Following the reviewer's suggestion, we have checked the entire manuscript and changed all instances of "buck density" to "bulk density" and "organic" to "organic matter". We have also carefully checked the use of "infiltration"

**Variations of deep soil moisture under different vegetation types and influencing factors in a watershed of the Loess Plateau, China**

**X. N. Fang[1], W. W. Zhao[1], L. X. Wang [2], Q. Feng [3], J. Y. Ding [1], Y. X. Liu [1] and X. Zhang [1]**

[1] State Key Laboratory of Earth Surface Processes and Resource Ecology, College of Resource and Technology, Beijing Normal University, Beijing 100875, P. R. China

[2] Department of Earth Sciences, Indiana University-Purdue University, Indianapolis (IUPUI), Indianapolis, Indiana 46202, USA

[3] College of Forestry, Shanxi Agricultural University, Taigu, Shanxi 030801, P. R. China

Correspondence to: W. W. Zhao (Zhaoww@bnu.edu.cn) and L. X. Wang (lxwang@iupui.edu)

**Abstract:**

Soil moisture in deep soil layers is a relatively stable water resource for vegetation growth in the semi-arid Loess Plateau of China. Characterizing the  variations of deep soil moisture and its influencing factors at a moderate watershed scale is important to ensure the sustainability of vegetation restoration efforts. In this study, we focused on analyzing the  variation and factors influencing deep soil moisture  (DSM) in (80-500 cm) soil layers based on a soil moisture survey of the Ansai watershed, Yanan, Shannxi province. Our results can be divided into four main findings. (1) At the watershed scale, the higher  variation of DSM occurred at  120-140 cm and 480-500 cm in the vertical direction. At a comparable depth but in the horizontal direction, the  variation of DSM under native vegetation was much lower than that in human-managed vegetation and introduced vegetation. (2) The DSM in native vegetation and human-managed vegetation was significantly higher than that of introduced vegetation, and different degrees of soil desiccation occurred under all introduced vegetation types. Among them, *Caragana korshinskii* and black locust caused most serious desiccation. (3) Taking the  DSM condition of native vegetation as a reference for local control,  DSM in this watershed could be divided into three layers: (Ⅰ) Rainfall transpiration layer (80-220cm). (Ⅱ) Transition layer (220-400cm). (Ⅲ) Stable layer (400-500 cm).~~Ⅰ) shallow rapid change layer (0-60 cm); Ⅱ) main rainfall infiltration layer (60-220 cm); Ⅲ) transition layer (220-400 cm); and Ⅳ) stable layer (400-500 cm). Positive and significant correlations existed between SMC at layers Ⅱ, Ⅲ and Ⅳ, and the correlations of the neighboring layer ranges were clearly stronger than that of nonadjacent depth ranges, although the SMC at shallow rapid change layer Ⅰ showed a disconnect (i.e., no correlations) with those at the three other soil depth layers.deep SMCland~~

management types. The main local controls of  DSM variation were soil particle composition and annual average rainfall; human agricultural management measures can alter soil bulk density, which contributes to higher DSM in farmland and apple orchard. In introduced vegetation, plant growth conditions, planting density, and litter water holding traits showed significant relationships with DSM. The results of this study are of practical significance for vegetation restoration strategies especially for the choice of vegetation types, planting zones, and proper human management measure.

**1 Introduction**

Soil moisture is an indispensable component of the terrestrial system and plays a critical role in surface hydrological processes, especially runoff generation, soil evaporation and plant transpiration (Cheema et al., 2011;Legates et al., 2010;Wang et al., 2012a;Zhao et al., 2013).  Soil moisture at different soil layers usually owns different hydrological processes and ecological function (Yang et al., 2012a). Surface or shallow layer soil moisture are usually greatly influenced by rainfall infiltration or evapotranspiration, and are regular water sources for vegetation growth, while moisture in deep soil layers works as a reservoir for soil. In rainy years DSM can be repleshed by rainfall infiltration, and in drought years it can also provide necessary water for plant growth. Thus, it is important for plant growth in dry seasons (Yang et al., 2012c;Jia and Shao, 2014).  This is particularly true in semi-arid areas, such as the Loess Plateau of China, where water resources are incredibly scarce. In such regions, DSM even becomes the main constraining factor of plant productivity and ecosystem sustainability (Wang et al., 2010c;Wang et al.,

2011a). In this study, we define deep soil layer as the layer whose soil moisture is not sensitive to daily evapotranspiration and regular rainfall event.

The Loess Plateau of China is located in a semi-arid area. The average annual rainfall in this region ranges from 150 to 800 mm, which is far lower than the average annual pan evaporation (1400–2000 mm) (Wang et al., 2010b). Low precipitation and high evaporation results in lower soil moisture content in this region. The shallow soil moisture is not sufficient to meet the needs of introduced vegetation growth (Yang et al., 2014b). Moreover, loess soil thickness in this area ranges from 30-80 m; at these depths, groundwater is not available for plants (Wang et al., 2013). Therefore, deep soil moisture,DSM which is stored in unsaturated soil, becomes an important water resource for plant growth (Yang et al., 2012c). However, the vegetation introduced by the national Grain for Green project tends to have strong water consumption. Large-scale afforestation has resulted in the excessive consumption of deep soil moistureDSM, and a large range of soil desiccation has been reported (Wang et al., 2008b;Wang et al., 2010b;Wang et al., 2010c;Wang et al., 2011b). Soil desiccation greatly reduces the capability of a "soil reservoir" to supply water to deep soil layers for plant growth in the Loess Plateau (Chen et al., 2008a). Introduced vegetation in desiccated land is easily degraded with low productivity, and "small aged tree" with a height of 3–5 m appeared widely. Therefore the sustainability of the restored ecosystem is being challenged. Moreover, traditional soil moisture studies, which have mainly focused on shallow depth layers (Baroni et al., 2013;BI et al., 2009;Gómez-Plaza et al., 2001), clearly cannot reveal the sustainability need for vegetation restoration.

Studies on deep soil moistureDSM have gradually drawn attention from many scientists in recent years. For example, it was recently found that deep soil moistureDSM was excessively consumed by almost all the introduced vegetation, and high planting density was the main reason for the severe deficit of soil moisture (Yang et al., 2012c). It was also found that introduced vegetation diminished the spatial heterogeneity of deep soil moistureDSM at the small catchment scale (Jia and Shao,

2014;Yang et al., 2014b). In recent years, several studies have been conducted on the variation and influencing factors of DSM in the Loess Plateau (Jia and Shao, 2014;Liu et al., 2010;Wang et al., 2013;Wang et al., 2012b;Yang et al., 2014a;Sun et al., 2014). Deep soil moisture is an indispensable water source for vegetation growth in the semi-arid Loess Plateau; understanding the variation and influencing factors of DSM is important for "timely, suitable, and moderate" vegetation restoration, and it can also help in developing proper measures that can help control soil desiccation. In fact, DSM is a result of long-term biophysical processes controlled by multiple factors (Vereecken et al., 2007). Several factors may impact DSM variation, such as vegetation traits, soil properties, topographical factors, climate factors, and human landscape management measures (Qiu et al., 2001;Zhu and Lin, 2011;Montenegro and Ragab, 2012;Vivoni et al., 2008;Lu et al., 2007;Lu et al., 2008). The dominant factors that affect DSM variation depend on the research scale (Entin et al., 2000). For instance, DSM variation was found to be mainly dominated by the type of vegetation at the slope scale (0.1-1 km$^2$) (Jia et al., 2013). It was also found that vegetation and topography are key factors contributing to DSM variation at the small catchment scale (1-100 km$^2$) (Yang et al., 2012a). Meanwhile, Wang et al. (2012b) reported that DSM variation at the regional scale (i.e., the  Loess Plateau, covering 640,000 km$^2$) is mainly determined by plant types and climatic conditions. Note that vegetation factors play an important role in the spatial variation of DSM at all scales (Western et al., 2004).

While all spatial scales, from slopes and small catchments to regions are relevant to the understanding of DSM variation, some scales are more operational and meaningful than others. For example, slopes and small catchments based studies tend to be too small in spatial extent to incorporate all environmental factors and human-managed measures (soil traits, climate characteristics, and human-managed measures in one slope or small catchment are usually homogeneous) that most relevant to DSM variation (Zhu et al., 2014a;BI et al., 2009;Zhu et al., 2014b;Gómez‑Plaza et al., 2000), whereas at the region scale, it is often impossible to assess essential mechanistic details (high variation of rainfall and temperature can cover the influencing effects of other factors) of DSM variation necessary for guiding local policies (Wang et al., 2010a;Wang et al., 2010d;Wang et al., 2012c). A moderate scale, covering an area of approximately 100-1000 km$^2$ over a watershed or a geopolitically-defined area represents a pivotal scale domain for the research of DSM variation mechanism. In particular, it is the scale at which people and nature mesh and interact most acutely (Zhao and Fang, 2014;Fang et al., 2015), and thus is a more operational scale for sustainable vegetation restoration policy making. Up to date, however, little particular research of  DSM variation has centered on such a moderate scale, and the variation mechanism of DSM at this kind of scale is still unclear.

In this study, we aimed to reveal the variation of DSM and its influencing factors at a moderate watershed scale.  This study included shallow root system vegetation and deep root system vegetation, covering eight specific vegetation types. We first identified the deep soil layer whose soil moisture is not sensitive to regular rainfall and daily evapotranspiration in Ansai watershed. Then we explored the overall variation of DSM in this area and compared the DSM of this two root system vegetation types as well as identify variations in their profiles. At last, the influence of various environmental factors on DSM under different vegetation types is discussed.

discussed. The objectives of this study were to: (1) quantify the variation characteristics of DSM; (2) explore the mechanisms for controlling DSM variability among different vegetation types at the watershed scale; (3) develop recommendations for land use management and the sustainability of vegetation recovery for the Loess Plateau.

**2 Materials and Methods**

**2.1 Study area**

The Yanhe watershed lies in the middle of the Loess Plateau in the northern Shaanxi Province. The Ansai watershed (108°47′-109°25′E, 36°52′-37°19′N) (Fig. 1) in this study is located in the upstream section of the Yanhe river, covering an area of approximately 1334 km$^2$, with a highly fragmented terrain; the elevation here ranges from 1057 m to 1743 m above sea level. This typical semi-arid loess hilly region has a mean annual temperature of 8.8 ℃ and  average annual precipitation  ranges from 375-546 mm across the watershed (Fig. S1). Most rainfall occurs in the form of thunderstorms during the summer months from June to September. Soil types in this study area include mainly loess soil with low fertility and vulnerability to soil erosion (Zhao et al., 2012) Soil texture is different across the watershed with sand content ranging from 24%-57%, slit content ranging from 40%-65%, and clay content ranging from 6%-10% (Fig. S2).

 The predominant land use types in Ansai watershed are rain-fed farmland, orchard land, sparse native grassland, pasture grassland, shrub land, and forest (Feng et al., 2013). The native vegetation in the study area consists of sparse grasses with shallow roots dominated by species, such as bunge needlegrass, common leymus, and Altai heterpappus. Non-native species, such as alfalfa, black locust, David peach, sea buckthorn, and *Caragana korshinskii*, were predominantly used in the study area under the national Grain for Green project. The cultivated crops are predominantly maize, millet and broom corn millet. Being in a semi-arid climatic zone, water resources represent the major constraint of vegetation growth and agricultural crop production.

[Figure]

Figure 1. Location of the study area and sampling sites.

**2.2 Sampling locations and description**

In this study, two vegetation groups of different root system were selected (1) shallow root system vegetation: native grasses (NG), farmland (FL) with human agricultural measures; (2) deep root system vegetation: pasture grasses (PG), sea buckthorn (SB), *Caragana korshinskii* (CK), David peach (DP), black locust (BL) and apple orchard (AO) with human agricultural measures. The description of the root distribution of selected vegetation types are provided in Table 1.

Table 1.The description of the root distribution of selected vegetation species.

[revised manuscript text omitted]

**3 Results**

**3.1 Deep soil moisture identification**

The soil moisture dynamic at 0-200cm during the sampling period are reported in Fig.2. As can be seen, the soil moisture in this two different root system vegetation fluctuates daily at 40 cm depth, while soil moisture at 80-200cm keeps constant with time going on. Thus, it can be concluded that the evapotranspiration during sampling period influences soil moisture no deeper than 80 cm in both shallow and deep root system vegetation. Combine Fig.2 and Fig.S3, the soil moisture in 40 cm does not change obviously with rainfall events which indicates the rainfall during monitoring time period influences soil moisture no deeper than 40cm.

[Figure]

Figure 2. Soil moisture (0-200cm) dynamic monitoring during the sampling period. Note: (a) native grasses with shallow root system, (b) *Caragana korshinskii* with deep root system.

According to Fig.S3, the mean air temperature and rainfall of dynamic monitoring time period in 2015 is similar to that of sampling time period in 2014, thus, we consider rainfall and evapotranspiration during the sampling time period influence soil moisture no deeper than 80cm. And in this study, we consider soil moisture in 80-500cm as deep soil moisture.

**3.2 Summary statistics of deep soil moisture**

The summary statistics of DSM at various depths are given in Table 3. In general, the mean soil moisture, SD, and CV were highly dependent on depth. The profile distributions of mean DSMt, SD, and CV are given in Table 3 and Fig. 3. The highest mean value (9.45%) was observed at the 400-500cm depth, the lowest (8.15%) was at the 120-140 cm depth, and the mean DSM below 300 cm was almost constant. However, both SD and CV showed waving trends with increasing depth (Fig. 3). The profile distributions of SD and CV were consistent. The highest values of both occurred at  100-120 cm and 480-500 cm (Table 3), which indicated that DSM at these depth ranges had relatively higher variability. Meanwhile, the lowest values occurred at  260-300 cm, which indicated lower variability of  DSM at these depth ranges. Most of the Kurtosis (expect for 80-120 cm) and Skewness values are positive, and the highest values of both occurred at 200-240cm depth. The Kolmogorov–Smirnov test indicated that soil moisture data sets were normally distributed. Thus, statistical analysis could be performed without data transformation (Shi et al., 2014).

Table 3. Summary statistics of deep soil moisture at various depths in the Ansai watershed.

[revised manuscript text omitted]

Deep SMC in native grassland zones is seldom affected by vegetation due to shallow root systems; thus, the deep SMC in native grasslands can be regarded as a reference for local control (Yang et al., 2012c). Based on the inflection point of DSM and the trending change of SD, the 80-500cm soil moisture profile of native grasslands can be divided into 3 layers: (1) 80-220 cm, at this layer, both DSM and SD at 80-220cm decreased as soil depth increased, which indicated that this layer may be a main rainfall infiltration layer. Furthermore, as depth increased, the level of rainfall infiltration decreased. (2) 220-400 cm, DSM in this layer remained relatively constant as soil depth increased, but its SD increased with soil depth, which indicated that this layer is unstable. We characterize it as a transition layer. (3) 400-500 cm, this is a relatively stable layer whose SD is constant as soil depth increases, despite increasing DSM with soil depth. At this layer, DSM is seldom influenced by rainfall infiltration. The profile distribution characteristics of farmland and apple orchards were similar to those of native grasslands, except for layer 300-500 cm. Perhaps this is because management measures increased the ranges of the rainfall infiltration layer. As for vegetation-introduced, DSM at 80-220cm depth of all vegetation types reached the lowest; at 220-500cm, the DSM of different introduced vegetation could be generally divided into three categories: (1) as soil depth increased, DSM increased (such as PG and SB); (2) as soil depth increased, DSM kept relative stable (such as DP and BL); (3) as soil depth increased, DSM increased first and then decreased (such as CK).

Based on the above analysis, we generally divided the DSM in this watershed into three layers: (Ⅰ) Rainfall transpiration layer (80-220cm). This layer is a main rainfall infiltration layer and can be greatly influenced by vegetation transpiration. (Ⅱ) Transition layer (220-400cm). This layer can be recharged by rainfall infiltration in rainy years, and can supply ordinary deep root vegetation with DSM in drought years (Ⅲ) Stable layer (400-500 cm). This is a relatively stable layer whose DSM is seldom influenced by rainfall infiltration in regular years, but can be influenced by extreme deep root vegetation such as CK and BL.

[revised manuscript text omitted]

---

## Referee Report (RR1)

COMMENTS TO THE REVISED VERSION OF THE MANUSCRIPT HESS-2016-22

Dear Editor,

I apologize for not being able to send my report before the deadline day.

The manuscript HESS-2016-22 was improved and reached a good quality level.

The authors gave a complete and detailed answer to all the comments to the previous version.

At the end, my suggestion is to accept the paper for publication.

Sincerely,

D.C.

---

## Author Response (AR2)

**Responses to Editor and Reviewers:**

**Responses to Editor**

***Editor Decision:*** *Publish subject to minor revisions (Editor review) (20 Jun 2016) by Prof. Nadia Ursino*

***Editor comments:*** *according to the reviewers' reports, your manuscript may be accepted for publication after revision. Try to address Referee#5 comments, and if you can, shorten your manuscript and upload the revised version.*

**Authors:** Dear Prof. Nadia Ursino, thank you so much for your decision and suggestion. We have carefully amended our manuscript based on the suggestions and comments, and tried our best to shorten the manuscript. The detailed responses are provided below.

**Responses to Reviewer #1:**

***Reviewer:*** *The quality of presentation has rooms to improve. For example, the numbers in the several tables have two decimal points. Are they really meaningful to have two, why not one?*

**Authors:** We agree that two decimal points are not meaningful in some tables. Thus, following the reviewer's suggestion, we have revised these tables (Table 2 on pages 8-9, Table 4 on page 18, Table 8 on pages 23-24).

***Reviewer:*** *I feel the tables are too excessive and it can be more concise to tell a story.*

**Authors:** Following the reviewer's suggestion, we adjusted some tables and moved some tables to supplement (Table 4 on page 18, Table 5-7 on pages 20-22, Table 8 on pages 23-24, Table S1-S3 in supplement).

***Reviewer:*** *The scale of Fig 2 in 0.01. This makes the fig rather busy. Does 0.01 really make a big difference?*

**Authors:** We agree that Fig. 2 looks a little busy due to the scale issue, and coarser-scale does not make a big difference. Therefore we adjusted the scale of Fig. 2 (Figure 2 on page 12).

**Response to reviewer #4:**

*Reviewer: The manuscript HESS-2016-22 was improved and reached a good quality level. The authors gave a complete and detailed answer to all the comments to the previous version. At the end, my suggestion is to accept the paper for publication.*

**Authors:** Thank you very much for your positive comments to our work on the revision of this manuscript.

**Response to reviewer #5:**

*Reviewer: This paper described the variations and controlling factors in deep soil moisture in the Ansai watershed in Yanan in Shanxi, China. The authors found that species-specific root system (vertical root distribution is a key factor in comparison with plant growth conditions, planting density and litter water holding traits controlling the variation in deep soil moisture. Generally, the paper is well written and address relevant scientific questions within the scope of HESS. However, I think the findings are not novel, but of more general known knowledge. The results presented are not robust. Therefore, I don't think the paper can be published in this form.*

**Authors:** Thank you so much for this comment. This comment drives us to think the manuscript over carefully. In our opinion, measuring deep soil moisture at depths of 0-500 cm is a very challenging and valuable task, there are several novel and interesting findings in the manuscript: (1) the main innovation of this study is to explore the variation of DSM and its influencing factors at a moderate watershed scale which is of great practical values for the ecological restoration policy in the Loess Plateau of China; (2) we clearly defined DSM based on our data analysis while we cannot find clear definition of DSM exists in previous studies; (3) we stratified the DSM based on its hydrology traits while almost all previous studies stratified the DSM subjectively; and (4)we found there existed significant difference in DSM under different introduced vegetation and identified the dominant influencing factors of DSM under different vegetation types. Therefore, we think our findings are not just known knowledge, and worth of publishing. We have further emphasized the above mentioned points.

*Reviewer: I would suggest to present CV of soil moistures along the depths vertically for each species. This analysis may help you understand the effect of species on deep soil moistures, and partition the effects of topography and species.*

**Authors:** We thank reviewer for this useful comments. We agree that the CV of DSM along the depths can really help us understand the effect of species on DSM, and partition the effects of topography. Thus, we have added the CV of DSM along the depths vertically in the revised manuscript (see lines 8-16 on page 17 and Figure S4 in supplement).

*Reviewer: The paper is not concisely written, but lengthy. Therefore, the paper needs to be shorten largely.*

**Authors:** Thank you so much for this comment. In order to make the manuscript more concise, we deleted some unnecessary contents throughout the manuscript and moved some tables to the supplement document (see lines 8-13 on page4, Table 5-7 on pages 20-22, lines 8-12 on page27, lines 20-22 on page27, lines 26-29 on page27, lines 15-18 on page28, lines 2-4 on page29, lines15-19 on page 29,lines19-20 on page31, lines 24-28 on page 31,lines 1-5 on page 32, lines 8-9 on page32, lines16-23 on page32).

*Reviewer: Conclusions is too long, need to be summarized.*

**Authors:** As suggested by the reviewer, we have shortened the conclusion section (see conclusion section on page 31-32).

[revised manuscript text omitted]